# I-Con: A Unifying Framework for Representation Learning

**Shaden Alshammari**[1]    **John Hershey**[2]    **Axel Feldmann**[1]    **William T. Freeman**[1,2]    **Mark Hamilton**[1,3]

[1] *MIT*    [2] *Google*    [3] *Microsoft*

https://aka.ms/i-con

Figure 1: **A "periodic" table of representation learning methods unified by the I-Con framework.** By choosing different types of conditional probability distributions over neighbors, I-Con generalizes over 23 commonly used representation learning methods.

## Abstract

As the field of representation learning grows, there has been a proliferation of different loss functions to solve different classes of problems. We introduce a single information-theoretic equation that generalizes a large collection of modern loss functions in machine learning. In particular, we introduce a framework that shows that several broad classes of machine learning methods are precisely minimizing an integrated KL divergence between two conditional distributions: the supervisory and learned representations. This viewpoint exposes a hidden information geometry underlying clustering, spectral methods, dimensionality reduction, contrastive learning, and supervised learning. This framework enables the development of new loss functions by combining successful techniques from across the literature. We not only present a wide array of proofs, connecting over 23 different approaches, but we also leverage these theoretical results to create state-of-the-art unsupervised image classifiers that achieve a +8% improvement over the prior state-of-the-art on unsupervised classification on ImageNet-1K. We also demonstrate that I-Con can be used to derive principled debiasing methods which improve contrastive representation learners.

## 1 Introduction

Over the past decade the field of representation learning has flourished, with new techniques, architectures, and loss functions emerging daily. These advances have powered state-of-the-art models in vision, language, and multimodal learning, often with minimal human supervision. Yet as the field

expands, the diversity of loss functions makes it increasingly difficult to understand how different methods relate, and which objectives are best suited for a given task.

In this work, we introduce a general mathematical framework that unifies a wide range of representation learning techniques spanning supervised, unsupervised, and self-supervised approaches under a single information-theoretic objective. Our framework, **Information Contrastive Learning (I-Con)**, reveals that many seemingly disparate methods including clustering, spectral graph theory, contrastive learning, dimensionality reduction, and supervised classification are all special cases of the same underlying loss function.

While prior work has identified isolated connections between subsets of representation learning methods, typically linking only two or three techniques at a time (Sobal et al., 2025; Hu et al., 2023; Yang et al., 2022; Böhm et al., 2023; Balestriero & LeCun, 2022), **I-Con is the first framework to unify over 23 distinct methods** under a single objective. This unified perspective not only clarifies the structure of existing techniques but also provides a strong foundation for transferring ideas and improvements across traditionally separate domains.

Using I-Con, we derive new unsupervised loss functions that significantly outperform previous methods on standard image classification benchmarks. Our key contributions are:

- We introduce *I-Con*, a single information-theoretic loss that generalizes several major classes of representation learning.
- We prove 15 theorems showing how diverse algorithms emerge as special cases of I-Con.
- We use I-Con to design a debiasing strategy that improves unsupervised ImageNet-1K accuracy by **+8%**, with additional gains of **+3%** on CIFAR-100 and **+2%** on STL-10 in linear probing.

## 2 RELATED WORK

Representation learning spans a wide range of methods for extracting structure from complex data. We review approaches that I-Con builds upon and generalizes. For comprehensive surveys, see (Le-Khac et al., 2020; Bengio et al., 2013; Weng, 2021).

**Feature Learning** aims to derive informative low-dimensional embeddings using supervisory signals such as pairwise similarities, nearest neighbors, augmentations, class labels, or reconstruction losses. Classical methods like PCA (Pearson, 1901) and MDS (Kruskal, 1964) preserve global structure, while UMAP (McInnes et al., 2018) and t-SNE (Hinton & Roweis, 2002; Van der Maaten & Hinton, 2008) focus on local topology by minimizing divergences between joint distributions. I-Con adopts a similar divergence-minimization view.

Contrastive learning approaches such as SimCLR (Chen et al., 2020a), CMC (Tian et al., 2020), CLIP (Radford et al., 2021), and MoCo v3 (Chen* et al., 2021) use positive and negative pairs, often built via augmentations or aligned modalities. I-Con generalizes these losses within a unified KL-based framework, highlighting subtle distinctions between them. Supervised classifiers (e.g., ImageNet models (Krizhevsky et al., 2017)) also yield effective features, which I-Con recovers by treating class labels as discrete contrastive points, bridging supervised and unsupervised learning.

**Clustering** methods uncover discrete structure through distance metrics, graph partitions, or contrastive supervision. Algorithms like k-Means (Macqueen, 1967), EM (Dempster et al., 1977), and spectral clustering (Shi & Malik, 2000) are foundational. Recent methods, including IIC (Ji et al., 2019), Contrastive Clustering (Li et al., 2021), and SCAN (Gansbeke et al., 2020), leverage invariance and neighborhood structure. Teacher-student models such as TEMI (Adaloglou et al., 2023) and EMA-based architectures (Chen et al., 2020b) enhance clustering further. I-Con encompasses these by aligning a clustering-induced joint distribution with a target distribution derived from similarity, structure, or contrastive signals.

**Unifying Representation Learning** has been explored through connections between contrastive learning and t-SNE (Hu et al., 2023; Böhm et al., 2023), equivalences between contrastive and cross-entropy losses (Yang et al., 2022), and relations between spectral and contrastive methods (Balestriero & LeCun, 2022; Sobal et al., 2025). Other efforts, like Bayesian grammar models (Grosse et al., 2012), offer probabilistic perspectives. Tschannen et al. (Tschannen et al., 2019) emphasized estimator and architecture design in mutual information frameworks but stopped short of broader unification.

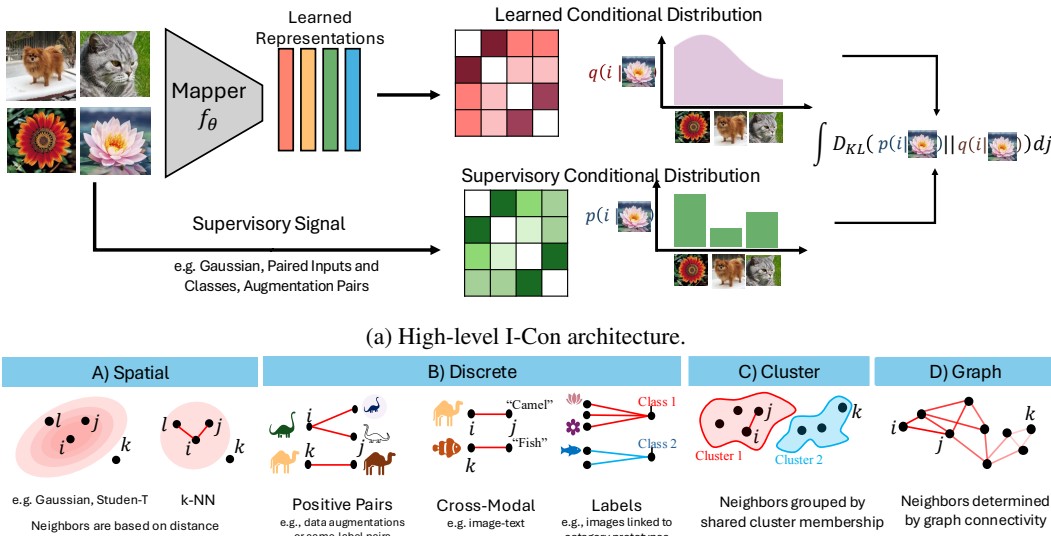

(a) High-level I-Con architecture.

(b) Illustrative examples of distribution families for $p_\theta$ or $q_\phi$.

Figure 2: **Overview of the I-Con framework**. (a) Alignment of learned and supervisory distributions. (b) Common distribution families in I-Con's formulation.

While prior work links subsets of these methods, I-Con, to our knowledge, is the first to unify supervised, contrastive, clustering, and dimensionality reduction objectives under a single loss. This perspective clarifies their shared structure and opens paths to new learning principles.

## 3 METHODS

The I-Con framework unifies multiple representation learning methods under a single loss function: minimizing the average KL divergence between two conditional "neighborhood distributions" that define transition probabilities between data points. This information-theoretic objective generalizes techniques from clustering, contrastive learning, dimensionality reduction, spectral graph theory, and supervised learning. By varying the construction of the supervisory distribution and the learned distribution, I-Con encompasses a broad class of existing and novel methods. We introduce I-Con and demonstrate its ability to unify techniques from diverse areas and orchestrate the transfer of ideas across different domains, leading to a state-of-the-art unsupervised image classification method.

### 3.1 INFORMATION CONTRASTIVE LEARNING

Let $i, j \in \mathcal{X}$ be elements of a dataset $\mathcal{X}$, with a probabilistic neighborhood function $p(j|i)$ defining a transition probability. To ensure valid probability distributions, $p(j|i) \geq 0$ and $\int_{j \in \mathcal{X}} p(j|i) = 1$. We parameterize this distribution by $\theta \in \Theta$, to create a learnable function $p_\theta(j|i)$. Similarly, we define another distribution $q_\phi(j|i)$ parameterized by $\phi \in \Phi$. The core I-Con loss function is then:

$$\mathcal{L}(\theta, \phi) = \int_{i \in \mathcal{X}} D_{\text{KL}}\left(p_\theta(\cdot|i) \| q_\phi(\cdot|i)\right) = \int_{i \in \mathcal{X}} \int_{j \in \mathcal{X}} p_\theta(j|i) \log \frac{p_\theta(j|i)}{q_\phi(j|i)}. \quad (1)$$

In practice, $p$ is typically a fixed "supervisory" distribution, while $q_\phi$ is learned by comparing deep network representations, prototypes, or clusters. Figure 2a illustrates this alignment process. The optimization aligns $q_\phi$ with $p$, minimizing their KL divergence. Although most existing methods optimize only $q_\phi$, I-Con also allows learning both $p_\theta$ and $q_\phi$, although one must take care to prevent trivial solutions.

## 3.2 UNIFYING REPRESENTATION LEARNING ALGORITHMS WITH I-CON

Despite the incredible simplicity of Equation 1, this equation is rich enough to generalize several existing methods in the literature simply by choosing parameterized neighborhood distributions $p_\theta$ and $q_\phi$ as shown in Figure 1. We categorize common choices for $p_\theta$ and $q_\phi$ in Figure 2a.

Table 1 summarizes some key choices which recreate popular methods from contrastive learning (SimCLR, MOCOv3, SupCon, CMC, CLIP, VICReg), dimensionality reduction (SNE, t-SNE, PCA), clustering (K-Means, Spectral, DCD, PMI), and supervised learning (Cross-Entropy and Harmonic Loss). Due to limited space, we defer proofs of each of these theorems to the supplemental material. We also note that Table 1 is not exhaustive, and we encourage the community to explore whether other learning frameworks implicitly minimize Equation 1 for some choice of $p$ and $q$.

### 3.2.1 EXAMPLE: SNE, SIMCLR, AND K-MEANS

While I-Con unifies a broad range of methods, we illustrate how different choices of $p$ and $q$ recover well-known techniques such as SNE, SimCLR, and K-Means. Full details are in the appendix.

**SNE as "neighbors remain neighbors."** Stochastic Neighbor Embedding (SNE) is a classic example. Given $x \in \mathbb{R}^{d \times n}$ with $n$ points in $d$ dimensions, SNE learns a low-dimensional representation $\phi \in \mathbb{R}^{m \times n}$, typically $m \ll d$. To preserve local structure, $p(j \mid i)$ is defined by placing a Gaussian around each high-dimensional point $x_i$, and $q_\phi(j \mid i)$ by placing a Gaussian around $\phi_i$. Minimizing the average KL divergence between these distributions ensures that points close in the original space remain close in the embedded space.

**SimCLR as "augmentations of the same image are neighbors."** Contrastive learning methods like SimCLR and SupCon instead use class labels. Here, $p(j \mid i) = 1$ if $j$ is an augmentation of $i$ (and 0 otherwise). In the embedding space, $q_\phi(j \mid i)$ is defined via a Gaussian-like distribution based on cosine similarity. Minimizing their KL divergence encourages images from the same scene to cluster together.

**K-Means as "points that are close are members of the same clusters."** Clustering-based approaches like K-Means and DCD follow a similar recipe. The distribution $p(j \mid i)$ is again Gaussian-based in the original space, while $q_\phi(j \mid i)$ reflects whether points are assigned to the same cluster in the learned representation. Minimizing KL divergence aligns these cluster assignments with the actual neighborhood structure in the data. Methods like K-Means include an entropy penalty to enforce hard probabilistic assignments, as shown in Theorem 13, whereas methods like DCD do not include it.

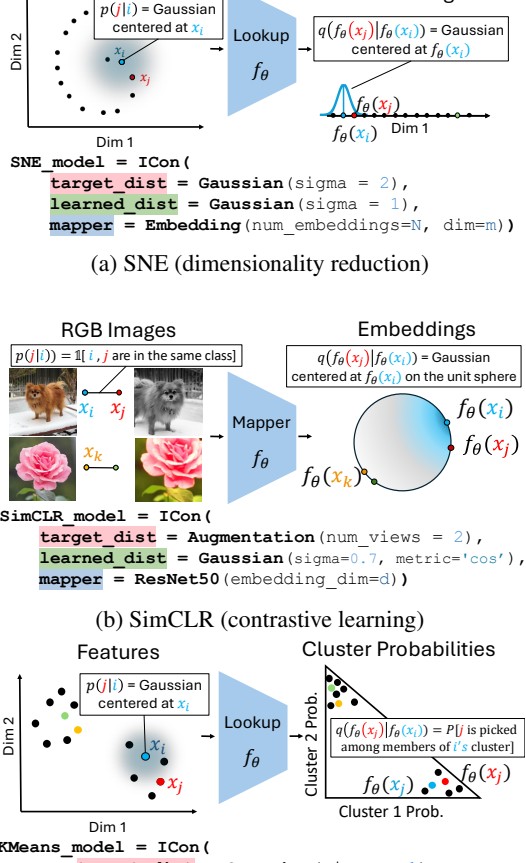

(a) SNE (dimensionality reduction)

(b) SimCLR (contrastive learning)

(c) K-Means (clustering)

Figure 3: Examples of methods as special cases of I-Con via different choices of $p$ and $q$, with corresponding code-style configurations.

## 3.3 CREATING NEW REPRESENTATION LEARNERS WITH I-CON

The I-Con framework unifies various approaches to representation learning under a single mathematical formulation and, crucially, facilitates the transfer of techniques among different domains.

| Method | Choice of $p_\theta(j \mid i)$ | Choice of $q_\phi(j \mid i)$ |
|---|---|---|
| **(A) Dimensionality Reduction** | | |
| **SNE** (Hinton & Roweis, 2002) Theorem 1 | Gaussian over data points, $x_i$ | Gaussian over learned low-dimensional points, $\phi_i$ $$\frac{\exp(-\|\phi_i - \phi_j\|^2)}{\sum_{k\neq i}\exp(-\|\phi_i - \phi_k\|^2)}$$ |
| **t-SNE** (Van der Maaten & Hinton, 2008) Corollary 1 | $$\frac{\exp(-\|x_i - x_j\|^2/2\sigma_i^2)}{\sum_{k\neq i}\exp(-\|x_i - x_k\|^2/2\sigma_i^2)}$$ | Cauchy distribution over $\phi_i$ $$\frac{(1+\|\phi_i - \phi_j\|^2)^{-1}}{\sum_{k\neq i}(1+\|\phi_i - \phi_k\|^2)^{-1}}$$ |
| **PCA** (Pearson, 1901) Theorem 2 | $\mathbb{1}[\,i=j\,]$ | Wide Gaussian on linear projection features, $f_\phi(x_i)$ $$\lim_{\sigma\to\infty}\frac{\exp(-\|f_\phi(x_i)-f_\phi(x_j)\|^2/2\sigma^2)}{\sum_{k\neq i}\exp(-\|f_\phi(x_i)-f_\phi(x_k)\|^2/2\sigma^2)}$$ |
| **(B) Contrastive Learning** | | |
| **InfoNCE Loss** (Bachman et al., 2019) Theorem 3 | $\frac{1}{Z}\mathbb{1}[i \text{ and } j \text{ are a positive pair}]$ | Gaussian on deep normalized features $$\frac{\exp\big(f_\phi(x_i)\cdot f_\phi(x_j)\big)}{\sum_{k\neq i}\exp\big(f_\phi(x_i)\cdot f_\phi(x_k)\big)}$$ |
| **Triplet Loss** (Schroff et al., 2015) Theorem 5 | | Gaussian on deep features (1 neg. sample, $\sigma\to 0$) $$\frac{\exp\big(-\|f_\phi(x_i)-f_\phi(x_j)\|^2/2\sigma^2\big)}{\sum_{k\in\{i^+,\,i^-\}}\exp\big(-\|f_\phi(x_i)-f_\phi(x_k)\|^2/2\sigma^2\big)}$$ |
| **t-SimCLR, t-SimCNE** (Hu et al., 2023; Böhm et al., 2023) Corollary 2 | | Student-T on deep features $$\frac{(1+\|\phi_i-\phi_j\|^2/\nu)^{-(\nu+1)/2}}{\sum_{k\neq i}(1+\|\phi_i-\phi_k\|^2/\nu)^{-(\nu+1)/2}}$$ |
| **VICReg\*** without covariance term (Bardes et al., 2021) Theorem 4 | | Wide Gaussian on learned features $$\lim_{\sigma\to\infty}\frac{\exp(-\|f_\phi(x_i)-f_\phi(x_j)\|^2/2\sigma^2)}{\sum_{k\neq i}\exp(-\|f_\phi(x_i)-f_\phi(x_k)\|^2/2\sigma^2)}$$ |
| **SupCon** (Khosla et al., 2020) Theorem 6 | $\frac{1}{Z}\mathbb{1}[i \text{ and } j \text{ have same class}]$ | |
| **X-Sample** (Sobal et al., 2025) Theorem 7 | Gaussian on corresponding embeddings $$\frac{\exp\big(g_\theta(x_i)\cdot g_\theta(x_j)\big)}{\sum_{k\neq i}\exp\big(g_\theta(x_i)\cdot_\theta(x_k)\big)}$$ | Gaussian on deep normalized features $$\frac{\exp\big(f_\phi(x_i)\cdot f_\phi(x_j)\big)}{\sum_{k\neq i}\exp\big(f_\phi(x_i)\cdot f_\phi(x_k)\big)}$$ |
| **LGSimCLR** (El Banani et al., 2023) | $\frac{1}{Z}\mathbb{1}[x_i \text{ is among } x_j\text{'s } k \text{ nearest neighbors}]$ | |
| **CMC & CLIP** (Tian et al., 2020) Theorem 8 | $\frac{1}{Z}\mathbb{1}[i,j \text{ pos. pairs}, V_i\neq V_j]$ | $$\frac{\exp\big(f_\phi(x_i)\cdot f_\phi(x_j)\big)}{\sum_{k\in V_j}\exp\big(f_\phi(x_i)\cdot f_\phi(x_k)\big)}$$ |
| **(C) Supervised Learning** | | |
| **Supervised Cross Entropy** (Good, 1963) Theorem 9 | Indicator over classes | $$\frac{\exp\big(f_\phi(x_i)\cdot \phi_j\big)}{\sum_{k\in C}\exp\big(f_\phi(x_i)\cdot \phi_k\big)}$$ |
| **Harmonic Loss** (Baek et al., 2025) Theorem 10 | $\mathbb{1}[i \text{ belongs to class } j]$ | Student-T on deep features and class prototypes $$\lim_{\sigma\to 0}\frac{(\sigma^2+\|f_\phi(x_i)-\phi_j\|^2)^{-n}}{\sum_{k\in C}(\sigma^2+\|f_\phi(x_i)-\phi_k\|)^{-n}}$$ |
| **Masked Lang. Modeling** (Devlin et al., 2019) Theorem 11 | $\frac{1}{Z}\,\#\big[\text{Context } i \text{ precedes token } j\big]$ | $$\frac{\exp\big(f_\phi(x_i)\cdot \phi_j\big)}{\sum_{k\in C}\exp\big(f_\phi(x_i)\cdot \phi_k\big)}$$ |
| **(D) Clustering** | | |
| **Probabilistic k-Means** (Macqueen, 1967) Theorem 13 | Intra-cluster uniform probability | Gaussians on datapoints $$\frac{\exp(-\|x_i-x_j\|^2/2\sigma_i^2)}{\sum_{k\neq i}\exp(-\|x_i-x_k\|^2/2\sigma_i^2)}$$ |
| **Spectral Clustering** (Ng et al., 2001) Corollary 4 | $$\sum_{c=1}^{m}\frac{p\big(f_\theta(x_i) \text{ and } f_\theta(x_j) \text{ in } c\big)}{\mathbb{E}[\text{size of cluster } c]}$$ | Gaussians on spectral embeddings $$\frac{\exp(-\|x_i-x_j\|^2/2\sigma_i^2)}{\sum_{k\neq i}\exp(-\|x_i-x_k\|^2/2\sigma_i^2)}$$ |
| **Normalized Cuts** (Shi & Malik, 2000) Theorem 14 | Intra-cluster uniform probability weighted by degree $$\sum_{c=1}^{m}\frac{p\big(f_\theta(x_i) \text{ and } f_\theta(x_j) \text{ in } c\big)\cdot d_j}{\mathbb{E}[\text{degree of cluster } c]}$$ | Gaussians on graph weigths $$\frac{\exp(w_{ij}/d_j)}{\sum_k \exp(w_{ik}/d_k)}$$ |
| **PMI Clustering** (Adaloglou et al., 2023) Theorem 15 | $\frac{1}{k}\mathbb{1}[j \text{ is } k\text{-NN of } i]$ | Intra-Cluster Uniform Probability $$\sum_{c=1}^{m}\frac{p\big(f_\theta(x_i) \text{ and } f_\theta(x_j) \text{ in } c\big)}{\mathbb{E}[\text{size of cluster } c]}$$ |
| **Debiased InfoNCE Clustering** (ours) | Debiased Graph through Uniform Distribution and Neighbor Propagation | $$\sum_{c=1}^{m}\frac{(1-\alpha)p\big(f_\theta(x_i) \text{ and } f_\theta(x_j) \text{ in } c\big)}{\mathbb{E}[\text{size of cluster } c]}+\frac{\alpha}{N}$$ |

Table 1: **I-Con unifies representation learners** under different choices of $p_\theta(j|i)$ and $q_\phi(j|i)$. Proofs of the propositions in this table can be found in the supplement.

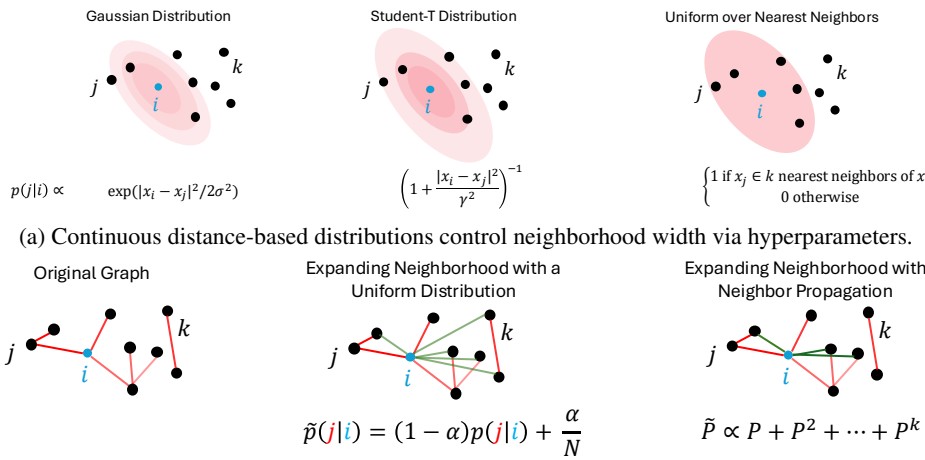

(a) Continuous distance-based distributions control neighborhood width via hyperparameters.

(b) Graph-based distributions expand neighborhoods through structural strategies.

Figure 4: **Neighborhood adaptation in continuous and discrete settings.** (a) Distance-based distributions modulate neighborhood "width" via parameters such as $\sigma$. (b) Graph-based approaches modify the connectivity directly, often via random walks or added edges, thereby broadening each node's neighborhood.

For instance, a trick from contrastive learning can be applied to clustering—or vice versa. In this paper, we demonstrate how surveying modern representation methods enables the development of clustering and unsupervised classification algorithms that surpass previous performance levels. Specifically, we integrate insights from spectral clustering, t-SNE, and debiased contrastive learning (Chuang et al., 2020) to build a state-of-the-art unsupervised image classification pipeline.

### 3.3.1   DEBASING

Debiased Contrastive Learning (DCL) addresses the mismatch caused by random negative sampling in contrastive learning, especially when the number of classes is small. Randomly chosen negatives can turn out to be positives, introducing spurious repulsive forces between similar examples. Chuang et al. (2020) rectify this by subtracting out such false repulsion terms and boosting attractive forces, substantially improving representation quality. However, their method modifies the softmax itself, implying that $q_{j|i}$ is no longer a genuine probability distribution and making it more difficult to extend the approach to clustering or supervised tasks.

Our view, grounded in the I-Con framework, suggests a simpler and more general alternative: rather than adjusting the learned distribution $q_{j|i}$, we incorporate additional "uncertainty" directly into the supervisory distribution $p(j|i)$. This preserves $q_{j|i}$ as a valid distribution and keeps the method applicable to tasks beyond contrastive learning.

### 3.3.2   DEBIASING THROUGH A UNIFORM DISTRIBUTION

Our first example adopts a simple uniform mixture:

$$\tilde{p}(j|i) \;=\; (1 - \alpha)\, p(j|i) \;+\; \frac{\alpha}{N},$$

where $N$ is the local neighborhood size, and $\alpha$ specifies the degree of mixing. This approach assigns a small probability mass $\frac{\alpha}{N}$ to each "negative" sample, thereby mitigating overconfident allocations. In supervised contexts, this is analogous to label smoothing (Szegedy et al., 2016). In contrast, Chuang et al. (2020) adjust the softmax function itself while retaining one-hot labels.

Another way to view this method is through the lens of heavier-tailed or broader distributions. By adding a uniform component, we mirror the idea in t-SNE's Student-$t$ distribution (Van der Maaten & Hinton, 2008), which allocates greater mass to distant points. In both cases, expanding the distribution reduces the chance of overfitting to a narrowly defined set of neighbors.

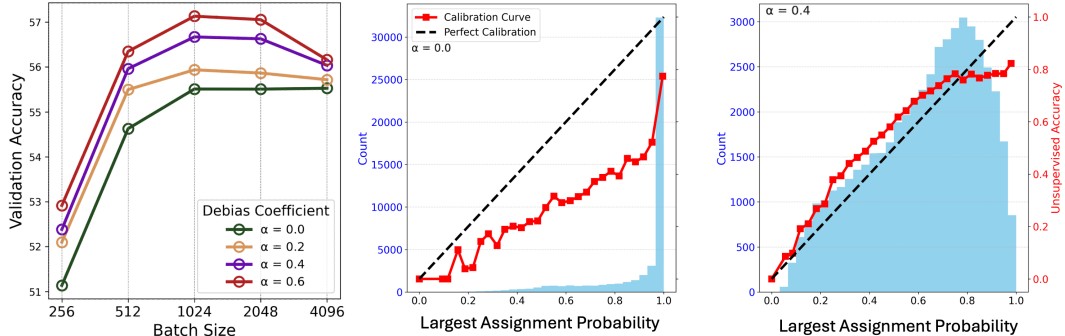

Figure 5: Left: Debiasing cluster learning improves performance on ImageNet-1K across batch sizes. Center: Distribution of maximum predicted probabilities for the biased model ($\alpha = 0$) showing poor calibration, with overconfident predictions. Right: Distribution of maximum predicted probabilities for the debiased model ($\alpha = 0.4$), demonstrating improved probability calibration. Debiased training alleviates optimization stiffness by reducing the prevalence of saturated logits, mitigating vanishing gradient issues, and fostering more robust and well-calibrated learning dynamics.

Empirical results in Tables 3, Figures 5, and 6 show that this lightweight modification consistently improves performance across various tasks and batch sizes. It also "relaxes" overconfident distributions, much like label smoothing in supervised cross entropy, thereby guarding against vanishing gradients.

### 3.3.3 DEBIASING THROUGH NEIGHBOR PROPAGATION

A second strategy applies graph-based expansions. As shown in Table 1, replacing k-Means' Gaussian neighborhoods with degree-weighted $k$-nearest neighbors recovers spectral clustering, which is known for robust, high-quality solutions. Building on this idea, we train contrastive learners with KNN-based neighborhood definitions. Given the nearest-neighbor graph, we can further expand it by taking longer walks, analogous to Word-Graph2Vec or tsNET (Li et al., 2023; Kruiger et al., 2017), a process we term *neighbor propagation*.

Formally, let $P$ be the conditional distribution matrix whose entries $P_{ij} = p(x_j \mid x_i)$ define the probability of selecting $x_j$ as a neighbor of $x_i$. Interpreting $P$ as the adjacency matrix of the training data, we can smooth it by summing powers of $P$ up to length $k$:

$$\tilde{P} \propto P + P^2 + \cdots + P^k.$$

We can further simplify this by taking a uniform distribution over all points reachable within $k$ steps, denoted by:

$$\tilde{P}_U \propto I\big[\, P + P^2 + \cdots + P^k \; > \; 0 \,\big],$$

where $I[\cdot]$ is the indicator function. This walk-based smoothing broadens the effective neighborhood, allowing the model to learn from a denser supervisory signal.

Tables 3 and 4 confirm that adopting such a propagation-based approach yields significant improvements in unsupervised image classification, underscoring the effectiveness of neighborhood expansion as a debiasing strategy.

## 4 EXPERIMENTS

In this section, we demonstrate that the I-Con framework offers testable hypotheses and practical insights into self-supervised and unsupervised learning. Rather than aiming only for state-of-the-art performance, our goal is to show how I-Con can enhance existing unsupervised learning methods

by leveraging a unified information-theoretic approach. Through this framework, we also highlight the potential for cross-pollination between techniques in varied machine learning domains, such as clustering, contrastive learning, and dimensionality reduction. This transfer of techniques, enabled by I-Con, can significantly improve existing methodologies and open new avenues for exploration.

We focus our experiments on clustering because it is relatively understudied compared to contrastive learning, and there are a variety of techniques that can now be adapted to this task. By connecting established methods such as k-Means, SimCLR, and t-SNE within the I-Con framework, we uncover a wide range of possibilities for improving clustering methods. We validate these theoretical insights experimentally, demonstrating the practical impact of I-Con.

We evaluate the I-Con framework using the ImageNet-1K dataset (Deng et al., 2009), which consists of 1,000 classes and over one million high-resolution images. This dataset is considered one of the most challenging benchmarks for unsupervised image classification due to its scale and complexity. To ensure a fair comparison with prior works, we strictly adhere to the experimental protocol introduced by (Adaloglou et al., 2023). The primary metric for evaluating clustering performance is Hungarian accuracy, which measures the quality of cluster assignments by finding the optimal alignment between predicted clusters and ground truth labels via the Hungarian algorithm (Ji et al., 2019). This approach provides a robust measure of clustering performance in an unsupervised context, where direct label supervision is absent during training.

For feature extraction, we utilize the DiNO pre-trained Vision Transformer (ViT) models in three variants: ViT-S/14, ViT-B/14, and ViT-L/14 (Caron et al., 2021). These models are chosen to ensure comparability with previous work and to explore how the I-Con framework performs across varying model capacities. The experimental setup, including training protocols, optimization strategies, and data augmentations, mirrors those used in TEMI to ensure consistency in methodology.

The training process involved optimizing a linear classifier on top of the features extracted by the DiNO models. Each model was trained for 30 epochs, using ADAM (Kingma & Ba, 2017) with a batch size of 4096 and an initial learning rate of 1e-3. We decayed the learning rate by a factor of 0.5 every 10 epochs to allow for stable convergence. We do not apply additional normalization to the feature vectors. During training, we applied a variety of data augmentation techniques, including random re-scaling, cropping, color jittering, and Gaussian blurring, to create robust feature representations. Furthermore, to enhance the clustering performance, we pre-computed global nearest neighbors for each image in the dataset using cosine similarity. This allowed us to sample two augmentations and two nearest neighbors for each image in every training batch, thus incorporating both local and global information into the learned representations. We refer to our derived approach as "InfoNCE Clusting" in Table 2. In particular, we use a supervisory neighborhood comprised of augmentations, KNNs ($k = 3$), and KNN walks of length 1. We use the "shared cluster likelihood by cluster" neighborhood from k-Means (See table 1 for a more detailed Equation) as our learned neighborhood function to drive cluster learning.

## 4.1 BASELINES

We compare our method against several state-of-the-art clustering methods, including TEMI, SCAN, IIC, and Contrastive Clustering. These methods rely on augmentations and learned representations, but often require additional regularization terms or loss adjustments, such as controlling cluster size or reducing the weight of affinity losses. In contrast, our I-Con-based loss function is self-balancing and does not require such manual tuning, making it a cleaner, more theoretically grounded approach. This allows us to achieve higher accuracy and more stable convergence across three different-sized backbones.

## 4.2 RESULTS

Table 2 compared the Hungarian accuracy of Debiased InfoNCE Clustering across different DiNO variants (ViT-S/14, ViT-B/14, ViT-L/14) and several other modern clustering methods. The I-Con framework consistently outperforms the prior state-of-the-art method across all model sizes. Specifically, for the DiNO ViT-B/14 and ViT-L/14 models, debiased InfoNCE clustering achieves significant performance gains of +4.5% and +7.8% in Hungarian accuracy compared to TEMI, the prior state-of-the-art ImageNet clusterer. We attribute these improvements to two main factors:

| Method | DiNO ViT-S/14 | DiNO ViT-B/14 | DiNO ViT-L/14 |
|---|---|---|---|
| k-Means | 51.84 | 52.26 | 53.36 |
| Contrastive Clustering | 47.35 | 55.64 | 59.84 |
| SCAN | 49.20 | 55.60 | 60.15 |
| TEMI | 56.84 | 58.62 | – |
| **Debiased InfoNCE Clustering (Ours)** | **57.8** $\pm$ 0.26 | **64.75** $\pm$ 0.18 | **67.52** $\pm$ 0.28 |

Table 2: Comparison of methods on ImageNet-1K clustering with respect to Hungarian Accuracy. Debiased InfoNCE Clustering significantly outperforms the prior state-of-the-art TEMI. Note that TEMI does not report results for ViT-L.

**Self-Balancing Loss:** Unlike TEMI or SCAN, which require hand-tuned regularizations (e.g., balancing cluster sizes or managing the weight of affinity losses), I-Con's loss function automatically balances these factors without additional regularization hyper-parameter tuning as we are using the exact same clustering kernel used by k-Means. This theoretical underpinning leads to more robust and accurate clusters.

**Cross-Domain Insights:** I-Con leverages insights from contrastive learning to refine clustering by looking at pairs of images based on their embeddings, treating augmentations and neighbors similarly. This approach, originally successful in contrastive learning, translates well into clustering and leads to improved performance on noisy high-dimensional image data.

### 4.3 ABLATIONS

We conduct several ablation studies to experimentally justify the architectural improvements that emerged from analyzing contrastive clustering through the I-Con framework. These ablations focus on two key areas: the effect of incorporating debiasing into the target and embedding spaces and the impact of neighbor propagation strategies.

| **Method** | DiNO ViT-S/14 | DiNO ViT-B/14 | DiNO ViT-L/14 |
|---|---|---|---|
| Baseline | 55.51 | 63.03 | 65.70 |
| + Debiasing | 57.27 $\pm$ 0.07 | 63.72 $\pm$ 0.09 | 66.87 $\pm$ 0.07 |
| + KNN Propagation | **58.45** $\pm$ 0.23 | 64.87 $\pm$ 0.19 | 67.25 $\pm$ 0.21 |
| + EMA | 57.8 $\pm$ 0.26 | **64.75** $\pm$ 0.18 | **67.52** $\pm$ 0.28 |

Table 3: Ablation study of new techniques discovered through the I-Con framework. We compare ImageNet-1K clustering accuracy across different sized backbones.

| **Method** | DiNO ViT-S/14 | DiNO ViT-B/14 | DiNO ViT-L/14 |
|---|---|---|---|
| Baseline | 55.51 | 63.03 | 65.72 |
| + KNNs | 56.43 | 64.26 | 65.70 |
| + 1-walks on KNN | **58.09** | **64.29** | 65.97 |
| + 2-walks on KNN | 57.84 | 64.27 | **67.26** |
| + 3-walks on KNN | 57.82 | 64.15 | 67.02 |

Table 4: Ablation Study on Neighbor Propagation. Adding both KNNs and walks of length 1 or 2 on the KNN graph achieves the best performance.

We perform experiments with different levels of debiasing in the target distribution, denoted by the parameter $\alpha$, and test configurations where debiasing is applied to the target side, both sides (target and learned representations), or none. As seen in Figure 6, adding debiasing improves performance, with the optimal value typically around $\alpha = 0.6$ to $\alpha = 0.8$, particularly when applied to both sides of the learning process. This method is similar to how debiasing work in contrastive learning by assuming that each negative sample has a non-zero probability ($\alpha/N$) of being incorrect. Figure 5 shows how changing the value of $\alpha$ improves performance across different batch sizes.

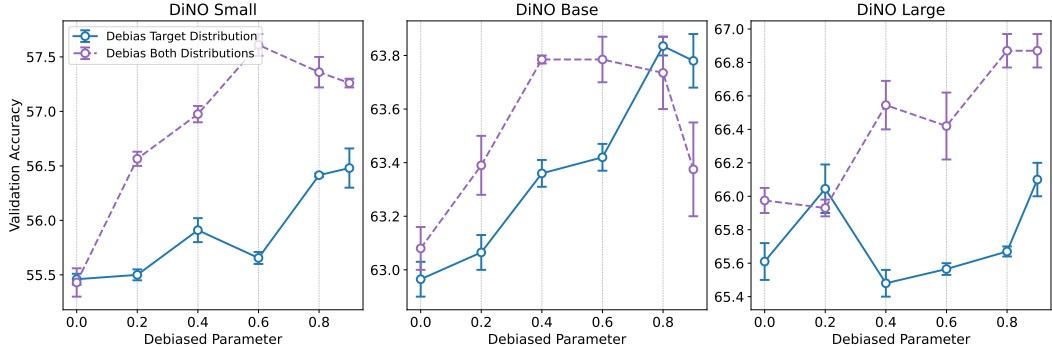

Figure 6: Effects of increasing the debias weight $\alpha$ on the supervisory neighborhood (blue line) and both the learned and supervisory neighborhood (red line). Adding some amount of debiasing helps in all cases, with a double debiasing yielding the largest improvements.

In a second set of experiments, shown in Table 4, we examine the impact of neighbor propagation strategies. We evaluate clustering performance when local and global neighbors are included in the contrastive loss computation. Neighbor propagation, especially at small scales ($s = 1$ and $s = 2$), significantly boosts performance across all model sizes, showing the importance of capturing local structure in the embedding space. Larger neighbor propagation values (e.g., $s = 3$) offer diminishing returns, suggesting that over-propagating neighbors may dilute the information from the nearest, most relevant points. Note that only DiNO-L/14 showed preference for large step size, and this is likely due to its higher k-nearest neighbor ability, so the augmented links are correct.

Our ablation studies highlight that small adjustments in the debiasing parameter and neighbor propagation can lead to notable improvements that achieve a state-of-the-art result with a simple loss function. Additionally, sensitivity to $\alpha$ and propagation size varies across models, with larger models generally benefiting more from increased propagation but requiring fine-tuning of $\alpha$ for optimal performance. We recommend using $\alpha \approx 0.6$ to $\alpha \approx 0.8$ and limiting neighbor propagation to small values for a balance between performance and computational efficiency.

## 5 CONCLUSION

In summary, we have developed I-Con: a single information-theoretic equation that unifies a broad class of machine learning methods. We provide over 15 theorems that prove this assertion for many of the most popular loss functions used in clustering, spectral graph theory, supervised and unsupervised contrastive learning, dimensionality reduction, and supervised classification and regression. We not only theoretically unify these algorithms but show that our connections can help us discover new state-of-the-art methods, and apply improvements discovered for a particular method to any other method in the class. We illustrate this by creating a new method for unsupervised image classification that achieves a +8% improvement over prior art. We believe that the results presented in this work represent just a fraction of the methods that are potentially unify-able with I-Con, and we hope the community can use this viewpoint to improve collaboration and analysis across algorithms and machine learning disciplines.

**Acknowledgments** This research was partially sponsored by the Department of the Air Force Artificial Intelligence Accelerator and was conducted under Cooperative Agreement Number FA8750-19-2-1000, as well as NSF CIF 1955864 (Occlusion and Directional Resolution in Computational Imaging). We also acknowledge support from Quanta Computer. Additionally, we would like thank Phillip Isola, Andrew Zisserman, Yair Weiss, Justin Kay, and Shivam Duggal for valuable discussions and feedback.

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

APPENDIX

## A    ADDITIONAL EXPERIMENTS ON DEBIASING FEATURE LEARNING

The following experiments aim to test the effect of our debiasing approach in feature learning. We followed the experimental setup introduced by Hu et al. (Hu et al., 2023). The architecture consisted of a ResNet-34 backbone paired with a two-layer multilayer perceptron (MLP) feature extractor. The MLP included a hidden layer with 512 units and an output layer with 64 units, without batch normalization.

**CIFAR-10 & CIFAR-100.** The models were trained on the CIFAR-10 dataset for 1000 epochs using the AdamW optimizer with the following hyperparameters: $\beta_1 = 0.9$, $\beta_2 = 0.999$, a learning rate of $1 \times 10^{-3}$, a batch size of 1024, and a weight decay of $1 \times 10^{-5}$. The learned kernel was either Gaussian or Student's t-distribution with degrees of freedom $d_f = 2$.

For evaluation, we used two methods: (1) linear probing on the 512-dimensional embeddings from the MLP's hidden layer, and (2) $k$-nearest neighbors ($k = 3$) classification based on the same embeddings for CIFAR-10 (in-distribution) and CIFAR-100 (out-of-distribution).

**STL-10 & Oxford-IIIT Pet.** With a similar setup, the models were trained contrastively on STL-10 (in distribution) without labels using the same hyperparameters as in the CIFAR experiments. For evaluation, we performed (1) linear probing for the STL-10 classification task and Oxford-IIIT Pet binary classification, and (2) $k$-nearest neighbors classification based on the same embeddings for STL-10 and Oxford-IIIT Pet with $k = 10$.

| Method | CIFAR10 (in distribution) | | CIFAR100 (out of distribution) | |
|---|---|---|---|---|
| | Linear Probing | KNN | Linear Probing | KNN |
| $q_\phi$ is a Gaussian Distribution | | | | |
| SimCLR (Chen et al., 2020a) | 77.79 | 80.02 | 31.82 | 40.27 |
| DCL (Chuang et al., 2020) | 78.32 | 83.11 | 32.44 | 42.10 |
| **Our Debiasing** $\alpha = 0.2$ | 79.50 | 84.07 | 32.53 | 43.19 |
| **Our Debiasing** $\alpha = 0.4$ | 79.07 | 85.06 | 32.53 | **43.29** |
| **Our Debiasing** $\alpha = 0.6$ | 79.32 | 85.90 | 30.67 | 29.79 |
| $q_\phi$ is a Student's t-distribution | | | | |
| t-SimCLR(Hu et al., 2023) | 90.97 | 88.14 | 38.96 | 30.75 |
| DCL (Chuang et al., 2020) | Diverges | Diverges | Diverges | Diverges |
| **Our Debiasing** $\alpha = 0.2$ | 91.31 | 88.34 | 41.62 | 32.88 |
| **Our Debiasing** $\alpha = 0.4$ | 92.70 | 88.50 | **41.98** | 34.26 |
| **Our Debiasing** $\alpha = 0.6$ | **92.86** | **88.92** | 38.92 | 32.51 |

Table 5: Contrastive feature learning evaluation results for CIFAR10 and CIFAR100 datasets with various debasing $\alpha$ factors. Adding some amount of debasing helps raising accuracy in both linear probing and KNN classification.

| Method | STL-10 (in distribution) | | Oxford-IIIT Pet (out of distribution) | |
|---|---|---|---|---|
| | Linear Probing | KNN | Logistic Regression | KNN |
| SimCLR (Chen et al., 2020a) | 77.71 | 74.92 | 74.80 | 71.48 |
| DCL (Chuang et al., 2020) | 78.32 | 75.03 | 74.41 | 70.22 |
| $q_\phi$ is a Student's t-distribution | | | | |
| t-SimCLR(Hu et al., 2023) | 85.11 | 83.05 | 83.40 | 81.41 |
| **Our Debiasing** $\alpha = 0.2$ | 85.94 | 83.15 | 84.11 | 81.15 |
| **Our Debiasing** $\alpha = 0.4$ | 86.13 | **84.14** | 84.07 | **84.13** |
| **Our Debiasing** $\alpha = 0.6$ | **87.18** | 83.58 | **84.51** | 83.04 |

Table 6: Contrastive feature learning evaluation results for STL10 (in distribution) and Oxford-IIIT Pet (out of distribution) with various debasing $\alpha$ factors. Similar to the other experiments, our debasing helps raising accuracy in both linear probing and KNN classification.

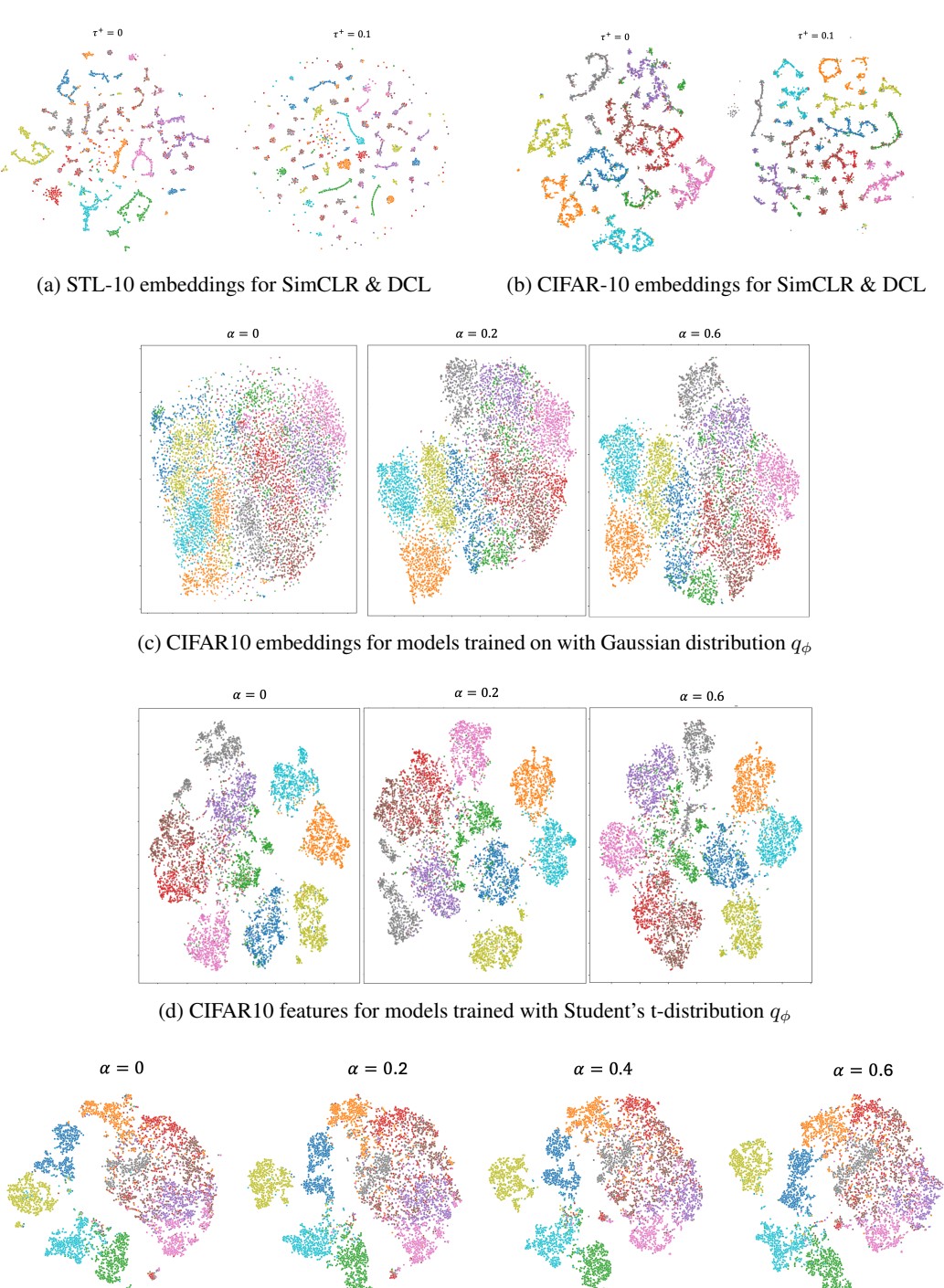

(a) STL-10 embeddings for SimCLR & DCL      (b) CIFAR-10 embeddings for SimCLR & DCL

(c) CIFAR10 embeddings for models trained on with Gaussian distribution $q_\phi$

(d) CIFAR10 features for models trained with Student's t-distribution $q_\phi$

(e) STL-10 features for models trained with Student's t-distribution $q_\phi$

Figure 7: t-SNE visualizations of learned embeddings on CIFAR10 and STL10 datasets. (a) and (b) display embeddings from the DCL (Chuang et al., 2020) method before and after applying debiasing, showing a tendency to heavily cluster data points, which may hinder out-of-distribution generalization (Hu et al., 2023). (c) and (d) show embeddings with Student's t-distribution, where the debiasing factor $\alpha$ enhances clustering and separation, resulting in improved data representation.

## B   PROOFS FOR UNIFYING DIMENSIONALITY REDUCTION METHODS

We begin by defining the setup for dimensionality reduction methods in the context of I-Con. Let $x_i \in \mathbb{R}^d$ represent high-dimensional data points, and $\phi_i \in \mathbb{R}^m$ represent their corresponding low-dimensional embeddings, where $m \ll d$. The goal of dimensionality reduction methods, such as Stochastic Neighbor Embedding (SNE) and t-Distributed Stochastic Neighbor Embedding (t-SNE), is to learn these embeddings such that neighborhood structures in the high-dimensional space are preserved in the low-dimensional space. In this context, the low-dimensional embeddings $\phi_i$ can be interpreted as the outputs of a mapping function $f_\theta(x_i)$, where $f_\theta$ is essentially an embedding matrix or look-up table. The I-Con framework is well-suited to express this relationship through a KL divergence loss between two neighborhood distributions: one in the high-dimensional space and one in the low-dimensional space.

**Theorem 1.** *Stochastic Neighbor Embedding (SNE)* ([Hinton & Roweis, 2002](#)) *is an instance of the I-Con framework.*

*Proof.* This is one of the most straightforward proofs in this paper, essentially based on the definition of SNE. The target distribution (supervised part), described by the neighborhood distribution in the high-dimensional space, is given by:

$$p_\theta(j|i) = \frac{\exp\left(-\|x_i - x_j\|^2 / 2\sigma_i^2\right)}{\sum_{k \neq i} \exp\left(-\|x_i - x_k\|^2 / 2\sigma_i^2\right)},$$

while the learned low-dimensional neighborhood distribution is:

$$q_\phi(j|i) = \frac{\exp\left(-\|\phi_i - \phi_j\|^2\right)}{\sum_{k \neq i} \exp\left(-\|\phi_i - \phi_k\|^2\right)}.$$

The objective is to minimize the KL divergence between these distributions:

$$\mathcal{L} = \sum_i D_{\mathrm{KL}}(p_\theta(\cdot|i) \| q_\phi(\cdot|i)) = \sum_i \sum_j p_\theta(j|i) \log \frac{p_\theta(j|i)}{q_\phi(j|i)}.$$

The embeddings $\theta_i$ are learned implicitly by minimizing $\mathcal{L}$. The mapper is an embedding matrix, as SNE is a non-parametric optimization. Therefore, SNE is a special case of the I-Con framework, where $p_\theta(j|i)$ and $q_\phi(j|i)$ represent the neighborhood probabilities in the high- and low-dimensional spaces, respectively. $\qquad\square$

**Corollary 1** (t-SNE ([Van der Maaten & Hinton](#), 2008)). *t-SNE is an instance of the I-Con framework.*

*Proof.* The proof is similar to the one for SNE. While the high-dimensional target distribution $p_\theta(j|i)$ remains unchanged, t-SNE modifies the low-dimensional distribution to a Student's t-distribution with one degree of freedom (Cauchy distribution):

$$q_\phi(j|i) = \frac{(1 + \|\phi_i - \phi_j\|^2)^{-1}}{\sum_{k \neq i}(1 + \|\phi_i - \phi_k\|^2)^{-1}}.$$

The objective remains to minimize the KL divergence. Therefore, t-SNE is an instance of the I-Con framework. $\qquad\square$

**Proposition 1.** *Let $X := \{x_i\}_{i=1}^n$, then the following cohesion variance loss*

$$\mathcal{L}_{cohesion\text{-}var} = \frac{1}{n} \sum_{ij} w_{ij} \|f_\phi(x_i) - f_\phi(x_j)\|^2 - 2Var(X)$$

*is an instance of $I - Con$ in the special case $w_{ij} = p(j|i)$ and $q_\phi$ is Gaussian as with a large width as $\sigma \to \infty$.*

*Proof.* By using AM-GM inequality, we have

$$\frac{1}{n}\sum_{k=1}^{n}e^{-z_k} \geq (\Pi_{k=1}^{n}e^{-z_k})^{\frac{1}{n}} \implies \frac{1}{n}\sum_{k=1}^{n}e^{-z_k} \geq (e^{-\sum_{k=1}^{n}z_k})^{\frac{1}{n}}$$

which implies that

$$\log\sum_{k=1}^{n}e^{-z_k} - \log n \geq \log\left(e^{-\sum_{k=1}^{n}z_k}\right)^{\frac{1}{n}} \implies \log\sum_{k=1}^{n}e^{-z_k} \geq -\frac{1}{n}\sum_{k=1}^{n}z_k + \log(n)$$

Alternatively, this can be written as

$$-\log\sum_{k=1}^{n}e^{-z_k} \leq \frac{1}{n}\sum_{k=1}^{n}z_k - \log(n)$$

Now assume that we have a Gaussian Kernel $q_\phi$

$$q_\phi(j|i) = \frac{\exp\left(-\|f_\phi(x_i) - f_\phi(x_j)\|^2/\sigma^2\right)}{\sum_{k\neq i}\exp\left(-\|f_\phi(x_i) - f_\phi(x_k)\|^2/\sigma^2\right)},$$

Therefore, given the inequality of exp-sum that we showed above, we have

$$\log q_\phi(j|i) = -\frac{\|f_\phi(x_i) - f_\phi(x_j)\|^2}{\sigma^2} - \log\sum_{k\neq i}\exp\left(-\frac{\|f_\phi(x_i) - f_\phi(x_k)\|^2}{\sigma^2}\right)$$

$$\leq -\frac{1}{\sigma^2}\|f_\phi(x_i) - f_\phi(x_j)\|^2 + \frac{1}{n\sigma^2}\sum_{k\neq i}\|f_\phi(x_i) - f_\phi(x_k)\|^2 - \log(n)$$

$$= -\frac{1}{\sigma^2}\left(-\|f_\phi(x_i) - f_\phi(x_j)\|^2 + \frac{1}{n}\sum_{k\neq i}\|f_\phi(x_i) - f_\phi(x_k)\|^2\right) - \log(n)$$

Therefore, the cross entropy $H(p_\theta, q_\phi)$, is bounded by

$$H(p_\theta, q_\phi) = -\frac{1}{n}\sum_i\sum_j p(j|i)\log q_\phi(j|i)$$

$$\leq \frac{1}{n}\sum_i\sum_j p(j|i)\left(\frac{1}{\sigma^2}(-\|f_\phi(x_i) - f_\phi(x_j)\|^2 + \frac{1}{n}\sum_{k\neq i}\|f_\phi(x_i) - f_\phi(x_k)\|^2) - \log(n)\right)$$

$$= \frac{1}{\sigma^2}\left(\frac{1}{n}\sum_{ij}p(j|i)\|f_\phi(x_i) - f_\phi(x_j)\|^2 - \frac{1}{n^2}\sum_{ijk}p(j|i)\|f_\phi(x_i) - f_\phi(x_k)\|^2\right) - \log(n)$$

$$= \frac{1}{\sigma^2}\left(\frac{1}{n}\sum_{ij}p(j|i)\|f_\phi(x_i) - f_\phi(x_j)\|^2 - 2\text{Var}(X)\right) + \log(n)$$

$$= \frac{1}{\sigma^2}\left(\frac{1}{n}\sum_{ij}p(j|i)\|f_\phi(x_i) - f_\phi(x_j)\|^2 - 2\text{Var}(X)\right) + \log(n)$$

$$= \frac{1}{\sigma^2}\mathcal{L}_{\text{cohesion-var}} + \log(n)$$

On the other hand, the L.H.S. can be upper bounded by using second order bound $e^{-z} \leq 1-z+z^2/2$, which implies that

$$-\log\sum_{k=1}^{n}e^{-z_k} \geq \log(1 - \frac{1}{n}\sum_{k=1}^{n}z_k + \frac{1}{n}\sum_{k=1}^{n}z_k^2) - \log(n)$$

On the other hand, $\log(1 + u) \geq u - u^2/2$, therefore,

$$-\log \sum_{k=1}^{n} e^{-z_k} \geq (1 - \frac{1}{n}\sum_{k=1}^{n} z_k + \frac{1}{n}\sum_{k=1}^{n} z_k^2) - \frac{1}{2}(1 - \frac{1}{n}\sum_{k=1}^{n} z_k + \frac{1}{n}\sum_{k=1}^{n} z_k^2)^2 - \log(n)$$

Therefore, in the limit $\sigma \to \infty$, the bounds become tighter and the I-Con loss approaches the cohesion variance loss. $\qquad \square$

**Theorem 2.** *Principal Component Analysis (PCA) is an asymptotic instance of the I-Con.*

*Proof.* By using Proposition 1. When $p_{j|i} = \mathbf{1}[i = j]$, we have the following expression for $\mathcal{L}$

$$\mathcal{L} = \frac{1}{n}\sum_{ij} p_{j|i}\|f_\phi(x_i) - f_\phi(x_j)\|^2 - 2\mathrm{Var}(X)$$

$$= \frac{1}{n}\sum_{i}\|f_\phi(x_i) - f_\phi(x_i)\|^2 - 2\mathrm{Var}(X)$$

$$= -2\mathrm{Var}(X)$$

Therefore, minimizing $\mathcal{L}$ is equivalent to maximizing the variance which is the equivalent of the PCA objective. Intuitivily, the KL divergence is asking $-\|f_\phi(x_i) - f_\phi(x_i)\|^2 = 0$ to be the maximum in comparison to $-\|f_\phi(x_i) - f_\phi(x_j)\|^2$ to match the supervisory indicator function, which implies the minimization of the sum of $-\|f_\phi(x_i) - f_\phi(x_j)\|^2$, which is maximizing the variance. If we restrict $f_\phi$ to be a linear projection map, then minimizing $\mathcal{L}$ would be equivalent to PCA. $\qquad \square$

## C  PROOFS FOR UNIFYING FEATURE LEARNING METHODS

We now extend the I-Con framework to feature learning methods commonly used in contrastive learning. Let $x_i \in \mathbb{R}^d$ be the input data points, and $f_\phi(x_i) \in \mathbb{R}^m$ be their learned feature embedding. In contrastive learning, the goal is to learn these embeddings such that similar data points (positive pairs) are close in the embedding space, while dissimilar points (negative pairs) are far apart. This setup can be expressed using a neighborhood distribution in the original space, where "neighbors" are defined not by proximity in Euclidean space, but by predefined relationships such as data augmentations or class membership. The learned embeddings $f_\phi(x_i)$ define a new distribution over neighbors, typically using a Gaussian kernel in the learned feature space. We show that InfoNCE is a natural instance of the I-Con framework, and many other methods, such as SupCon, CMC, and Cross Entropy, follow from this.

**Theorem 3** (InfoNCE (Bachman et al., 2019))**.** *InfoNCE is an instance of the I-Con framework.*

*Proof.* InfoNCE aims to maximize the similarity between positive pairs while minimizing it for negative pairs in the learned feature space. In the I-Con framework, this can be interpreted as minimizing the divergence between two distributions: the neighborhood distribution in the original space and the learned distribution in the embedding space.

The neighborhood distribution $p_\theta(j|i)$ is uniform over the positive pairs, defined as:

$$p_\theta(j|i) = \begin{cases} \frac{1}{k} & \text{if } x_j \text{ is among the } k \text{ positive views of } x_i, \\ 0 & \text{otherwise.} \end{cases}$$

where $k$ is the number of positive pairs for $x_i$.

The learned distribution $q_\phi(j|i)$ is based on the similarities between the embeddings $f_\phi(x_i)$ and $f_\phi(x_j)$, constrained to unit norm ($\|f_\phi(x_i)\| = 1$). Using a temperature-scaled Gaussian kernel, this distribution is given by:

$$q_\phi(j|i) = \frac{\exp\left(f_\phi(x_i) \cdot f_\phi(x_j)/\tau\right)}{\sum_{k\neq i}\exp\left(f_\phi(x_i) \cdot f_\phi(x_k)/\tau\right)},$$

where $\tau$ is the temperature parameter controlling the sharpness of the distribution. Since $\|f_\phi(x_i)\| = 1$, the Euclidean distance between $f_\phi(x_i)$ and $f_\phi(x_j)$ is $2 - 2(f_\phi(x_i) \cdot f_\phi(x_j))$.

The InfoNCE loss can be written in its standard form:

$$\mathcal{L}_{\text{InfoNCE}} = -\sum_i \log \frac{\exp\left(f_\phi(x_i) \cdot f_\phi(x_i^+)/\tau\right)}{\sum_k \exp\left(f_\phi(x_i) \cdot f_\phi(x_k)/\tau\right)},$$

where $j^+$ is the index of a positive pair for $i$. Alternatively, in terms of cross-entropy, the loss becomes:

$$\mathcal{L}_{\text{InfoNCE}} \propto \sum_i \sum_j p_\theta(j|i) \log q_\phi(j|i) = H(p_\theta, q_\phi),$$

where $H(p_\theta, q_\phi)$ denotes the cross-entropy between the two distributions. Since $p_\theta(j|i)$ is fixed, minimizing the cross-entropy $H(p_\theta, q_\phi)$ is equivalent to minimizing the KL divergence $D_{\text{KL}}(p_\theta \| q_\phi)$. By aligning the learned distribution $q_\phi(j|i)$ with the target distribution $p_\theta(j|i)$, InfoNCE operates within the I-Con framework, where the neighborhood structure in the original space is preserved in the embedding space. Thus, InfoNCE is a direct instance of I-Con, optimizing the same divergence-based objective. $\qquad \square$

**Corollary 2.** *t-SimCLR and t-SimCNE (Hu et al., 2023; Böhm et al., 2023) are instances of the I-Con framework.*

Given the proof of Theorem 3, we can see that t-SimCLR is equivelant by having the same $p_\theta$ but $q_\phi$ would change from a Gaussian distribution over cosine similarity to a Student-T distribution over a Euclidean distance.

$$q_\phi(j|i) = \frac{\left(\|f_\phi(x_i) - f_\phi(x_j)\|^2/\tau\right)^{-1}}{\sum_{k \neq i}\left(\|f_\phi(x_i) - f_\phi(x_k)\|^2/\tau\right)^{-1}},$$

**Theorem 4.** *VICReg Bardes et al. (2021) without a covariance term is an instance of the I-Con framework.*

Given Proposition 1, we know that any loss in the cohesion variance form is an instance of $I$-Con:

$$\mathcal{L} = \frac{1}{n}\sum_{ij} p_{j|i}\|f_\phi(x_i) - f_\phi(x_j)\|^2 - 2\text{Var}(X)$$

If we choose $p_{j|i}$ to be an indicator over positive pairs, $i$ and $i^+$, we obtain

$$\mathcal{L} = \frac{1}{n}\sum_i \|f_\phi(x_i) - f_\phi(x_{i^+})\|^2 - 2\text{Var}(X)$$

which is the VICReg loss without the covariance term and with an invariance-to-variance term ratio of 1:2. Observe that VICReg does not have negative pairs because it applies an equal repulsion force to all points. This is equivalent to taking $\sigma \to \infty$ in the conditional Gaussian distribution over the embeddings.

**Theorem 5** (Triplet Loss (Schroff et al., 2015))**.** *Triplet Loss can be viewed as an instance of the I-Con framework with the following distributions $p_\theta(j|i)$ and $q_\phi(j|i)$:*

$$p_\theta(j|i) = \begin{cases} \frac{1}{k} & \text{if } x_j \text{ is among the } k \text{ positive views of } x_i, \\ 0 & \text{otherwise,} \end{cases}$$

$$q_\phi(j|i) = \frac{\exp\left(-\frac{\|f_\phi(x_i) - f_\phi(x_j)\|^2}{\sigma^2}\right)}{\sum_{k \neq i} \exp\left(-\frac{\|f_\phi(x_i) - f_\phi(x_k)\|^2}{\sigma^2}\right)},$$

*particularly in the special case where only two neighbors are considered: one positive view and one negative view.*

*Proof.* The idea of this proof was first presented at (Khosla et al., 2020) using Taylor Approximation; however, in this proof we present a stronger bounds for this result. For simplicity, we set $\sigma = 1$ (the general bounds for other $\sigma$ values are provided at the end of the proof).

$$\mathcal{L} = -\frac{1}{N} \sum_i \sum_j q_\phi(j|i) \log \frac{\exp\left(-\|f_\phi(x_i) - f_\phi(x_j)\|^2\right)}{\sum_{k \neq i} \exp\left(-\|f_\phi(x_i) - f_\phi(x_k)\|^2\right)}.$$

In the special case where each anchor $x_i$ has exactly one positive $x_i^+$ and one negative $x_i^-$ example, the denominator simplifies to:

$$\sum_{k \neq i} \exp\left(-\|f_\phi(x_i) - f_\phi(x_k)\|^2\right) = \exp\left(-\|f_\phi(x_i) - f_\phi(x_i^+)\|^2\right) + \exp\left(-\|f_\phi(x_i) - f_\phi(x_i^-)\|^2\right).$$

Let $d_i^+ = \|f_\phi(x_i) - f_\phi(x_i^+)\|^2$ and $d_i^- = \|f_\phi(x_i) - f_\phi(x_i^-)\|^2$. Substituting these into the loss function, we obtain:

$$\mathcal{L} = -\frac{1}{N} \sum_i \log \frac{\exp\left(-d_i^+\right)}{\exp\left(-d_i^+\right) + \exp\left(-d_i^-\right)}$$

$$= -\frac{1}{N} \sum_i \log \left(\frac{1}{1 + \exp\left(d_i^- - d_i^+\right)}\right)$$

$$= \frac{1}{N} \sum_i \log \left(1 + \exp\left(d_i^+ - d_i^-\right)\right).$$

Recognizing that the expression inside the logarithm is the softplus function, we can leverage its well-known bounds:

$$\max(z, 0) \leq \log\left(1 + \exp(z)\right) \leq \max(z, 0) + \log(2).$$

By letting $z = d_i^+ - d_i^-$, we substitute into the bounds to obtain:

$$\frac{1}{N} \sum_i \max(d_i^+ - d_i^-, 0) \leq \mathcal{L} \leq \frac{1}{N} \sum_i \max(d_i^+ - d_i^-, 0) + \log(2),$$

where the left-hand side is the Triplet loss $\mathcal{L}_{\text{Triplet}} = \frac{1}{N} \sum_i \max(d_i^+ - d_i^-, 0)$. Therefore, we obtain the following bounds:

$$\mathcal{L} - \log(2) \leq \mathcal{L}_{\text{Triplet}} \leq \mathcal{L}.$$

For a general $\sigma$, the inequality bounds are as follows:

$$\mathcal{L}_\sigma - \sigma^2 \log(2) \leq \mathcal{L}_{\text{Triplet}} \leq \mathcal{L}_\sigma,$$

where

$$\mathcal{L}_\sigma = -\frac{\sigma^2}{N} \sum_i \sum_j q_\phi(j|i) \log \frac{\exp\left(-\frac{\|f_\phi(x_i) - f_\phi(x_j)\|^2}{\sigma^2}\right)}{\sum_{k \neq i} \exp\left(-\frac{\|f_\phi(x_i) - f_\phi(x_k)\|^2}{\sigma^2}\right)}.$$

As $\sigma$ approaches 0, $\mathcal{L}_{\text{Triplet}}$ approaches $\mathcal{L}_\sigma$. $\qquad\square$

**Theorem 6.** *The Supervised Contrastive Loss (Khosla et al., 2020) is an instance of the I-Con framework.*

*Proof.* This follows directly from Theorem 3. Define the supervisory and target distributions as:

$$q_\phi(j \mid i) = \frac{\exp\left(f_\phi(x_i) \cdot f_\phi(x_j)/\tau\right)}{\sum_{k \neq i} \exp\left(f_\phi(x_i) \cdot f_\phi(x_k)/\tau\right)},$$

$$p_\theta(j \mid i) = \frac{1}{K_i - 1} \mathbf{1}[i \text{ and } j \text{ share the same label}],$$

where $f_\phi$ is the mapping to deep feature space and $K_i$ is the number of samples in the class of $i$. Substituting these definitions into the I-Con framework recovers the Supervised Contrastive Loss. $\qquad\square$

**Theorem 7.** *The X-Sample Contrastive Learning Loss (Sobal et al., 2025) is an instance of the I-Con framework.*

*Proof.* Consier the following $p$ distribution over corresponding features (e.g. caption embeddings for images):

$$\frac{\exp\big(g_\theta(x_i) \cdot g_\theta(x_j)\big)}{\sum\limits_{k \neq i} \exp\big(g_\theta(x_i) \cdot_\theta (x_k)\big)}$$

where $g$ could be either a parametric or a non-parametric mapper to the corresponding embeddings $g_\theta(x_i)$. On the other hand, similar to most feature learning methods, the learned distribution is a Gaussian over learned embeddings with cosine distance

$$q_\phi(j \mid i) = \frac{\exp\big(f_\phi(x_i) \cdot f_\phi(x_j)\big)}{\sum\limits_{k \neq i} \exp\big(f_\phi(x_i) \cdot f_\phi(x_k)\big)}$$

where $f_\phi$ is the mapping to deep feature space. $\square$

**Theorem 8.** *Contrastive Multiview Coding (CMC) and CLIP are instances of the I-Con framework.*

*Proof.* Since we have already established that InfoNCE is an instance of the I-Con framework, this corollary follows naturally. The key difference in Contrastive Multiview Coding (CMC) and CLIP is that they optimize alignment across different modalities. The target probability distribution $p_\theta(j|i)$ can be expressed as:

$$p_\theta(j|i) = \frac{1}{Z} \mathbb{1}[i \text{ and } j \text{ are positive pairs and } V_i \neq V_j],$$

where $V_i$ and $V_j$ represent the modality sets of $x_i$ and $x_j$, respectively. Here, $p_\theta(j|i)$ assigns uniform probability over positive pairs drawn from different modalities.

The learned distribution $q_\phi(j|i)$, in this case, is based on a Gaussian similarity between deep features, but conditioned on points from the opposite modality set. Thus, the learned distribution is defined as:

$$q_\phi(j|i) = \frac{\exp\left(-\|f_\phi(x_i) - f_\phi(x_j)\|^2\right)}{\sum_{k \in V_j} \exp\left(-\|f_\phi(x_i) - f_\phi(x_k)\|^2\right)}.$$

This formulation shows that CMC and CLIP follow the same principles as InfoNCE but apply them in a multiview setting, fitting seamlessly within the I-Con framework by minimizing the divergence between the target and learned distributions across different modalities. $\square$

**Theorem 9.** *Cross-Entropy classification is an instance of the I-Con framework.*

*Proof.* Cross-Entropy can be viewed as a special case of the CMC loss, where one "view" corresponds to the data point features and the other to the class logits. The affinity between a data point and a class is based on whether the point belongs to that class. This interpretation has been explored in prior work, where Cross-Entropy was shown to be related to the CLIP loss (Yang et al., 2022). $\square$

**Theorem 10.** *Harmonic Loss for classification is an instance of the I-Con framework.*

*Proof.* This is the equivalent of moving from a Gaussian distribution for $q(j|i)$ in Cross-Entropy to a Student-T distribution analogs to moving from SNE to t-SNE. More specifically, let $V$ be the set of data points, $C$ the set of class prototypes, $\phi_i$ be the learned class prototype for class $i$, and $n$ be the harmonic loss degree.

Consider the following $p$, which is a data-label indicator

$$p(j|i) = \mathbb{1}\big[i \text{ belongs to class } j\big]$$

and the following $q$, which is a Student-T distribution with $2n - 1$ degrees for freedom.

$$\lim_{\sigma \to 0} \frac{(1 + \|f_\phi(x_i) - \phi_j\|^2/((2n-1)\sigma^2))^{-n}}{\sum_{k \in C}(1 + \|f_\phi(x_i) - \phi_k\|^2/((2n-1)\sigma^2))^{-n}}$$

It can be rewritten as

$$\lim_{\sigma \to 0} \frac{(((2n-1)\sigma^2) + \|f_\phi(x_i) - \phi_j\|^2)^{-n}}{\sum_{k \in C}(((2n-1)\sigma^2) + \|f_\phi(x_i) - \phi_k\|^2/)^{-n}}$$

As $\sigma \to \infty$, the loss function approaches

$$\mathcal{L} = \sum_{i \in C} \frac{(\|f_\phi(x_i) - \phi_j\|^2)^{-n}}{\sum_{k \in C}(\|f_\phi(x_i) - \phi_k\|^2/)^{-n}}$$

which's the Harmonic Loss for classification as introduced by 10 □

**Theorem 11.** *Masked Language Modeling (MLM) (Devlin et al., 2019) loss is an instance of the I-Con framework.*

*Proof.* In Masked Language Modeling, the objective is to predict a masked token $j$ given its surrounding context $x_i$. This setup fits naturally within the I-Con framework by defining appropriate target and learned distributions.

The target distribution $p_\theta(j|i)$ is the empirical distribution over contexts $i$ and tokens $j$, defined as:

$$p_\theta(j|i) = \frac{1}{Z}\#\left[\text{Context } i \text{ precedes token } j\right],$$

where $\#\left[\text{Context } i \text{ precedes token } j\right]$ counts the number of times token $j$ follows context $x_i$ in the training corpus and $Z$ is a normalization constant ensuring that $\sum_j p_\theta(j|i) = 1$.

The learned distribution $q_\phi(j|i)$ is modeled using the neural network's output logits for token predictions. It is defined as a softmax over the dot product between the context embedding $f_\phi(x_i)$ and the token embeddings $\phi_j$:

$$q_\phi(j|i) = \frac{\exp\left(f_\phi(x_i) \cdot \phi_j\right)}{\sum_{k \in \mathcal{V}} \exp\left(f_\phi(x_i) \cdot \phi_k\right)},$$

where $f_\phi(x_i)$ is the embedding of the context $x_i$ produced by the model, $\phi_j$ is the embedding of token $j$, and $\mathcal{V}$ is the vocabulary of all possible tokens.

The MLM loss aims to minimize the cross-entropy between the target distribution $p_\theta(j|i)$ and the learned distribution $q_\phi(j|i)$:

$$\mathcal{L}_{\text{MLM}} = -\sum_i \sum_j p_\theta(j|i) \log q_\phi(j|i) = H(p_\theta, q_\phi).$$

Since in practice, for each context $x_i$, only the true masked token $j_i^*$ is considered, the target distribution simplifies to:

$$p_\theta(j|i) = \delta_{j,j_i^*},$$

where $\delta_{j,j_i^*}$ is the Kronecker delta function, equal to 1 if $j = j_i^*$ and 0 otherwise.

Substituting this into the loss function, the MLM loss becomes:

$$\mathcal{L}_{\text{MLM}} = -\sum_i \log q_\phi(j_i^*|x_i).$$

□

## D   PROOFS FOR UNIFYING CLUSTERING METHODS

The connections between clustering and the I-Con framework are more intricate compared to the dimensionality reduction methods discussed earlier. To establish these links, we first introduce a probabilistic formulation of K-means and demonstrate its equivalence to the classical K-means algorithm, showing that it is a zero-gap relaxation. Building upon this, we reveal how probabilistic K-means can be viewed as an instance of I-Con, leading to a novel clustering kernel. Finally, we show that several clustering methods implicitly approximate and optimize for this kernel.

**Definition 1** (Classical K-means). *Let $x_1, x_2, \ldots, x_N \in \mathbb{R}^n$ denote the data points, and $\mu_1, \mu_2, \ldots, \mu_m \in \mathbb{R}^n$ be the cluster centers.*

*The objective of classical K-means is to minimize the following loss function:*

$$\mathcal{L}_{k\text{-}Means} = \sum_{i=1}^{N} \sum_{c=1}^{m} \mathbb{1}(c^{(i)} = c) \|x_i - \mu_c\|^2,$$

*where $c^{(i)}$ represents the cluster assignment for data point $x_i$, and is defined as:*

$$c^{(i)} = \arg\min_c \|x_i - \mu_c\|^2.$$

PROBABILISTIC K-MEANS RELAXATION

In probabilistic K-means, the cluster assignments are relaxed by assuming that each data point $x_i$ belongs to a cluster $c$ with probability $\phi_{ic}$. In other words, $\phi_i$ represents the cluster assignments vector for $x_i$

**Proposition 2.** *The relaxed loss function for probabilistic K-means is given by:*

$$\mathcal{L}_{Prob\text{-}k\text{-}Means} = \sum_{i=1}^{N} \sum_{c=1}^{m} \phi_{ic} \|x_i - \mu_c\|^2,$$

*and is equivalent to the original K-means objective $\mathcal{L}_{k\text{-}Means}$. The optimal assignment probabilities $\phi_{ic}$ are deterministic, assigning probability 1 to the closest cluster and 0 to others.*

*Proof.* For each data point $x_i$, the term $\sum_{c=1}^{m} \phi_{ic} \|x_i - \mu_c\|^2$ is minimized when the assignment probabilities $\phi_{ic}$ are deterministic, i.e.,

$$\phi_{ic} = \begin{cases} 1 & \text{if } c = \arg\min_j \|x_i - \mu_j\|^2, \\ 0 & \text{otherwise.} \end{cases}$$

With these deterministic probabilities, $\mathcal{L}_{\text{Prob-k-Means}}$ simplifies to the classical K-means objective, confirming that the relaxation introduces no gap. □

CONTRASTIVE FORMULATION OF PROBABILISTIC K-MEANS

**Definition 2.** *Let $\{x_i\}_{i=1}^{N}$ be a set of data points. Define the conditional probablity $q_\phi(j|i)$ as:*

$$q_\phi(j|i) = \sum_{c=1}^{m} \frac{\phi_{ic} \phi_{jc}}{\sum_{k=1}^{N} \phi_{kc}},$$

*where $\phi_i$ is the soft-cluster assignments for $x_i$.*

Given $q_\phi(j|i)$, we can reformulate probabilistic K-means as a contrastive loss:

**Theorem 12.** *Let $\{x_i\}_{i=1}^{N} \in \mathbb{R}^n$ and $\{\phi_{ic}\}_{i=1}^{N}$ be the corresponding assignment probabilities. Define the objective function $\mathcal{L}$ as:*

$$\mathcal{L} = -\sum_{i,j} (x_i \cdot x_j) \, q_\phi(j|i).$$

*Minimizing $\mathcal{L}$ with respect to the assignment probabilities $\{\phi_{ic}\}$ yields optimal cluster assignments equivalent to those obtained by K-means.*

*Proof.* The relaxed probabilistic K-means objective $\mathcal{L}_{\text{Prob-k-Means}}$ is:

$$\mathcal{L}_{\text{Prob-k-Means}} = \sum_{i=1}^{N} \sum_{c=1}^{m} \phi_{ic} \|x_i - \mu_c\|^2.$$

Expanding this, we obtain:

$$\mathcal{L}_{\text{Prob-k-Means}} = \sum_{c=1}^{m} \left( \sum_{i=1}^{N} \phi_{ic} \right) \|\mu_c\|^2 - 2 \sum_{c=1}^{m} \left( \sum_{i=1}^{N} \phi_{ic} x_i \right) \cdot \mu_c + \sum_{i=1}^{N} \|x_i\|^2.$$

The cluster centers $\mu_c$ that minimize this loss are given by:

$$\mu_c = \frac{\sum_{i=1}^{N} \phi_{ic} x_i}{\sum_{i=1}^{N} \phi_{ic}}.$$

Substituting $\mu_c$ back into the loss function, we get:

$$\mathcal{L} = -\sum_{i,j} (x_i \cdot x_j) \, q_\phi(j|i),$$

which proves that minimizing this contrastive formulation leads to the same clustering assignments as classical K-means. □

**Corollary 3.** *The alternative loss function:*

$$\mathcal{L} = -\sum_{i,j} \|x_i - x_j\|^2 \, q_\phi(j|i),$$

*yields the same optimal clustering assignments when minimized with respect to $\{\phi_{ic}\}$.*

*Proof.* Expanding the squared norm in the loss function gives:

$$\mathcal{L} = -\sum_{i,j} \left( \|x_i\|^2 - 2x_i \cdot x_j + \|x_j\|^2 \right) q_\phi(j|i).$$

The terms involving $\|x_i\|^2$ and $\|x_j\|^2$ simplify since $\sum_j q_\phi(j|i) = 1$, reducing the loss to:

$$\mathcal{L} = 2 \left( -\sum_{i,j} x_i \cdot x_j q_\phi(j|i) \right),$$

which is equivalent to the objective in the previous theorem. □

PROBABILISTIC K-MEANS AS AN I-CON METHOD

In the I-Con framework, the target and learned distributions represent affinities between data points based on specific measures. For instance, in SNE, these measures are Euclidean distances in high- and low-dimensional spaces, while in SupCon, the distances reflect whether data points belong to the same class. Similarly, we can define a measure of neighborhood probabilities in the context of clustering, where two points are considered neighbors if they belong to the same cluster. The probability of selecting $x_j$ as $x_i$'s neighbor is the probability that a point, chosen uniformly at random from $x_i$'s cluster, is $x_j$. More explicitly, let $q_\phi(j|i)$ represent the probability that $x_j$ is selected uniformly at random from $x_i$'s cluster:

$$q_\phi(j|i) = \sum_{c=1}^{m} \frac{\phi_{ic} \phi_{jc}}{\sum_{k=1}^{N} \phi_{kc}}.$$

**Theorem 13** (K-means as an instance of I-Con). *Given data points $\{x_i\}_{i=1}^{N}$, define the neighborhood probabilities $p_\theta(j|i)$ and $q_\phi(j|i)$ as:*

$$p_\theta(j|i) = \frac{\exp\left(-\|x_i - x_j\|^2 / 2\sigma^2\right)}{\sum_k \exp\left(-\|x_i - x_k\|^2 / 2\sigma^2\right)}, \quad q_\phi(j|i) = \sum_{c=1}^{m} \frac{\phi_{ic} \phi_{jc}}{\sum_{k=1}^{N} \phi_{kc}}.$$

*Let the loss function $\mathcal{L}_{c\text{-}SNE}$ be the sum of KL divergences between the distributions $q_\phi(j|i)$ and $p_\theta(j|i)$:*

$$\mathcal{L}_{c\text{-}SNE} = \sum_i D_{KL}(q_\phi(\cdot|i)\|p_\theta(\cdot|i)).$$

*Then,*

$$\mathcal{L}_{c\text{-}SNE} = \frac{1}{2\sigma^2}\mathcal{L}_{Prob\text{-}k\text{-}Means} - \sum_i H(q_\phi(\cdot|i)),$$

*where $H(q_\phi(\cdot|i))$ is the entropy of $q_\phi(\cdot|i)$.*

*Proof.* For simplicity, assume that $2\sigma^2 = 1$. Denote $\sum_k \exp\left(-\|x_i - x_k\|^2\right)$ by $Z_i$. Then we have:

$$\log p_\theta(j|i) = -\|x_i - x_j\|^2 - \log Z_i.$$

Let $\mathcal{L}_i$ be defined as $-\sum_j \|x_i - x_j\|^2 q_\phi(j|i)$. Using the equation above, $\mathcal{L}_i$ can be rewritten as:

$$\mathcal{L}_i = -\sum_j \|x_i - x_j\|^2 q_\phi(j|i) \tag{2}$$

$$= \sum_j (\log(p_\theta(j|i)) + \log(Z_i))q_\phi(j|i) \tag{3}$$

$$= \sum_j q_\phi(j|i)\log(p_\theta(j|i)) + \sum_j q_\phi(j|i)\log(Z_i) \tag{4}$$

$$= \sum_j q_\phi(j|i)\log(p_\theta(j|i)) + \log(Z_i) \tag{5}$$

$$= H(q_\phi(\cdot|i), p_\theta(\cdot|i)) + \log(Z_i) \tag{6}$$

$$= D_{\text{KL}}(q_\phi(\cdot|i)\|p_\theta(\cdot|i)) + H(q_\phi(\cdot|i)) + \log(Z_i). \tag{7}$$

Therefore, $\mathcal{L}_{\text{Prob-KMeans}}$, as defined in Corollary 3, can be rewritten as:

$$\mathcal{L}_{\text{Prob-KMeans}} = -\sum_i \sum_j \|x_i - x_j\|^2 q_\phi(j|i) = \sum_i \mathcal{L}_i \tag{8}$$

$$= \sum_i D_{\text{KL}}(q_\phi(\cdot|i)\|p_\theta(\cdot|i)) + H(q_\phi(\cdot|i)) + \log(Z_i) \tag{9}$$

$$= \mathcal{L}_{\text{c-SNE}} + \sum_i H(q_\phi(\cdot|i)) + \text{constant}. \tag{10}$$

Therefore,

$$\mathcal{L}_{\text{c-SNE}} = \mathcal{L}_{\text{Prob-KMeans}} - \sum_i H(q_\phi(\cdot|i)).$$

If we allow $\sigma$ to take any value, the entropy penalty will be weighted accordingly:

$$\mathcal{L}_{\text{c-SNE}} = \frac{1}{2\sigma^2}\mathcal{L}_{\text{Prob-KMeans}} - \sum_i H(q_\phi(\cdot|i)).$$

Note that the relation above is up to an additive constant. This implies that minimizing the contrastive probabilistic K-means loss with entropy regularization minimizes the sum of KL divergences between $q_\phi(\cdot|i)$ and $p_\theta(\cdot|i)$. □

**Corollary 4.** *Spectral Clustering is an instance of the I-Con framework.*

*Proof.* From Theorem 13, we know that K-Means clustering can be formulated as an instance of the I-Con framework, where the clustering assignments depend on the inner products of the data points.

Spectral Clustering extends this idea by first embedding the data into a lower-dimensional space using the top $k$ eigenvectors of the normalized Laplacian derived from the affinity matrix $A$. The affinity matrix $A$ is constructed using a similarity measure (e.g., an RBF kernel) and encodes the probabilities of assignments between data points. Given this transformation, spectral clustering is an instance of I-Con on the projected embeddings. □

**Theorem 14.** *Normalized Cuts ([Shi & Malik, 2000](#)) is an instance of I-Con.*

*Proof.* The proof for this follows naturally from our work on K-Means analysis. The loss function for normalized cuts is defined as:

$$\mathcal{L}_{\text{NormCuts}} = \sum_{c=1}^{m} \frac{\text{cut}(A_c, \overline{A}_c)}{\text{vol}(A_c)},$$

where $A_c$ is a subset of the data corresponding to cluster $c$, $\overline{A}_c$ is its complement, and $\text{cut}(A_c, \overline{A}_c)$ represents the sum of edge weights between $A_c$ and $\overline{A}_c$, while $\text{vol}(A_c)$ is the total volume of cluster $A_c$, i.e., the sum of edge weights within $A_c$.

Similar to K-Means, by reformulating this in a contrastive style with soft-assignments, the learned distribution can be expressed using the probabilistic cluster assignments $\phi_{ic} = p(c|x_i)$ as:

$$q_\phi(j|i) = \sum_{c=1}^{m} \frac{\phi_{ic}\phi_{jc}d_j}{\sum_{k=1}^{N} \phi_{kc}d_k},$$

where $d_j$ is the degree of node $x_j$, and the volume and cut terms can be viewed as weighted sums over the soft-assignments of data points to clusters.

This reformulation shows that normalized cuts can be written in a manner consistent with the I-Con framework, where the target distribution $p_\theta(j|i)$ and the learned distribution $q_\phi(j|i)$ represent affinity relationships based on graph structure and cluster assignments.

Thus, normalized cuts is an instance of I-Con, where the loss function optimizes the neighborhood structure based on the cut and volume of clusters in a manner similar to K-Means and its probabilistic relaxations. □

**Theorem 15.** *Mutual Information Clustering is an instance of I-Con.*

*Proof.* Given the connection established between SimCLR, K-Means, and the I-Con framework, this result follows naturally. Specifically, the target distribution $p_\theta(j|i)$ (the supervised part) is a uniform distribution over observed positive pairs:

$$p_\theta(j|i) = \begin{cases} \frac{1}{k} & \text{if } x_j \text{ is among the } k \text{ positive views of } x_i, \\ 0 & \text{otherwise.} \end{cases}$$

On the other hand, the learned embeddings $\phi_i$ represent the probabilistic assignments of $x_i$ into clusters. Therefore, similar to the analysis of the K-Means connection, the learned distribution is modeled as:

$$q_\phi(j|i) = \sum_{c=1}^{m} \frac{\phi_{ic}\phi_{jc}}{\sum_{k=1}^{N} \phi_{kc}}.$$

This shows that Mutual Information Clustering can be viewed as a method within the I-Con framework, where the learned distribution $q_\phi(j|i)$ aligns with the target distribution $p_\theta(j|i)$, completing the proof. □

## E   I-CON AS A VARIATIONAL METHOD

Variational bounds for mutual information are widely explored and have been connected to loss functions such as InfoNCE, where minimizing InfoNCE maximizes the mutual information lower bound ([Oord et al., 2018](#); [Poole et al., 2019](#)). The proof usually starts by rewriting the mutual information:

$$I(X;Y) = \mathbb{E}_{p(x,y)}\left[\log\frac{q(x|y)}{p(x)}\right] + \mathbb{E}_{p(y)}\left[D_{\text{KL}}\left(p(x|y) \,\|\, q(x|y)\right)\right]$$

This expression is typically used to derive a lower bound for $I(X;Y)$. The proof usually begins by assuming that $p$ is uniform over discrete data points $\mathcal{X} = \{x_i\}_{i=1}^{N}$ (i.e., we use uniform sampling

for data points). By using the fact that $p(x_i) = \frac{1}{N}$, we can write $p(x, y) = \frac{1}{N} p(x|y)$. Therefore, the mutual information lower bound becomes

$$
\begin{aligned}
I(X; Y) &\geq \mathbb{E}_{p(x,y)} \left[ \log q(x|y) \right] - \mathbb{E}_{p(x,y)} \left[ \log p(x) \right] \\
&= \mathbb{E}_{p(x,y)} \left[ \log q(x|y) \right] + \log(N) \\
&= \frac{1}{N} \sum_{x,y \in \mathcal{X} \times \mathcal{X}} p(x|y) \log q(x|y) + \log(N) \\
&= \frac{1}{N} \sum_{y \in \mathcal{X}} \sum_{x \in \mathcal{X}} p(x|y) \log q(x|y) + \log(N) \\
&= -H\left(p(x|y), q(x|y)\right) + \log(N)
\end{aligned}
$$

Therefore, maximizing the cross-entropy between the two distributions maximizes the mutual information between samples.

On the hand, Variational Bayesian (VB) methods are fundamental in approximating intractable posterior distributions $p(z \mid x)$ with tractable variational distributions $q_\phi(z)$. This approximation is achieved by minimizing the Kullback-Leibler (KL) divergence between the variational distribution and the true posterior:

$$
\mathrm{KL}(q_\phi(z) \| p(z \mid x)) = \mathbb{E}_{q_\phi(z)} \left[ \log \frac{q_\phi(z)}{p(z \mid x)} \right]. \tag{11}
$$

The optimization objective, known as the Evidence Lower Bound (ELBO), is given by:

$$
\mathrm{ELBO} = \mathbb{E}_{q_\phi(z)} \left[ \log p(x, z) \right] - \mathbb{E}_{q_\phi(z)} \left[ \log q_\phi(z) \right]. \tag{12}
$$

Maximizing the ELBO is equivalent to minimizing the KL divergence, thereby ensuring that $q_\phi(z)$ closely approximates $p(z \mid x)$ (Blei et al., 2017).

VB can be framed within the I-Con framework by making specific mappings between the variables and distributions. Let $i$ correspond to the data point $x$, and $j$ correspond to the latent variable $z$. We can set the supervisory distribution $p_\theta(z \mid x)$ to be the true posterior $p(z \mid x)$. This allow us to define the learned distribution $q_\phi(z \mid x)$ to be independent of $x$, i.e., $q_\phi(z \mid x) = q_\phi(z)$.

Under these settings, the I-Con loss simplifies to:

$$
\mathcal{L}(\phi) = \int_{x \in \mathcal{X}} \mathrm{KL}\left(p(z \mid x) \| q_\phi(z)\right) \, dx = \mathbb{E}_{p(x)} \left[ \mathrm{KL}(p(z \mid x) \| q_\phi(z)) \right]. \tag{13}
$$

INTERPRETATION

- Global Approximation: In VB, $q_\phi(z)$ serves as a global approximation to the posterior $p(z \mid x)$ across all data points $x$. Similarly, in I-Con, when $q_\phi(j \mid i) = q_\phi(j)$, the learned distribution provides a uniform approximation across all $i$.

- Variational Alignment: Both frameworks employ variational techniques to align a tractable distribution $q_\phi$ with an intractable or supervisory distribution $p$. This alignment ensures that the learned representations capture essential information from the target distribution.

- Framework Generalization: I-Con generalizes VB by allowing $q_\phi(j \mid i)$ to depend on $i$, enabling more flexible and data-specific alignments. VB is recovered as a special case where the learned distribution is uniform across all data points.

## F   WHY DO WE NEED TO UNIFY REPRESENTATION LEARNERS?

I-con not only provides a deeper understanding of these methods but also opens up the possibility of creating new methods by mixing and matching components. We explicitly use this property to discover new improvements to both clustering and representation learners. In short, I-Con acts like a periodic table of machine learning losses. With this periodic table we can more clearly see the implicit assumptions of each method by breaking down modern ML losses into more simple components: pairwise conditional distributions $p$ and $q$.

One particular example of how this opens new possibilities is with our generalized debiasing operation. Through our experiments we show adding a slight constant linkage between datapoints improves both stability and performance across clustering and feature learning. Unlike prior art, which only applies to specific feature learners, our debiasers can improve clusterers, feature learners, spectral graph methods, and dimensionality reducers.

Finally it allows us to discover novel theoretical connections by compositionally exploring the space, and considering limiting conditions. We use I-Con to help derive a novel theoretical equivalences between K-Means and contrastive learning, and between MDS, PCA, and SNE. Transferring ideas between methods is standard in research, but in our view it becomes much simpler to do this if you know methods are equivalent. Previously, it might not be clear how exactly to translate an insight like changing Gaussian distributions to Cauchy distributions in the upgrade from SNE to T-SNE has any effect on clustering or representation learning. In I-Con it becomes clear to see that similarly softening clustering and representation learning distributions can improve performance and debias representations.

## G   HOW TO CHOOSE NEIGHBORHOOD DISTRIBUTIONS FOR YOUR PROBLEM

PARAMETERIZATION OF LEARNING SIGNAL

- **Parametric**: (Learn a network to transform a data points to representations). Use a parametric method to quickly represent new datapoints without retraining. Use a parametric method if there is enough "features" in the underlying data to properly learn a representation. Use this option with datasets with sparse supervisory signal in order to share learning signal through network parameters.

- **Nonparametric**: (Learn one representation per data point). Use a nonparametric method if datapoints are abstract and don't contain natural features that are useful for mapping. Use this option to better optimize the loss of each individual datapoint. Do not use this in sparse supervisory signal regimes (Like augmentation based contrastive learning), as there are not enough links to resolve each individual embedding.

CHOICE OF SUPERVISORY SIGNAL

- **Gaussians on distances in the input space**: though this is a common choice and underlies methods like k-means, with enough data it is almost always better to use k-neighbor distributions as they better capture local topology of data. This is the same intuition that is used to justify spectral clustering over k-means.

- **K-neighbor graphs distributions**: If your data can be naturally put into a graph instead of just considering Gaussians on the input space we suggest it. This allows the algorithm to adapt local neighborhoods to the data, as opposed to considering all points neighborhoods equally shaped and sized. This better aligns with the manifold hypothesis.

- **Contrastive augmentations**: When possible, add contrastive augmentations to your graph - this will improve performance in cases where quantities of interest (like an image class) are guaranteed to be shared between augmentations.

- **General kernel smoothing techniques**: Use random walks to improve the optimization quality. It connects more points together and in some cases mirrors geodesic distance on the manifold (Crane et al., 2013).

- **Debiasing**: Use this if you think negative pairs actually have a small chance of aligning positively. For a small number of classes this parameter scales like the inverse of the number of classes. You can also use this to improve stability of the optimization.

CHOICE OF REPRESENTATION:

Any conditional distribution on representations can be used, so consider what kind of structure you want to learn, tree, vector, cluster, etc. And choose the distribution to be simple and meaningful for that representation.

- **Discrete**: Use discrete cluster-based representations if interpretability and discrete structure are important
- **Continuous Vector**: Use a vector representation if generic downstream performance is a concern as this is a bit easier to optimize than discrete variants.

## H  COMPARING I-CON, MLE, AND THE KL DIVERGENCE

There are many connections between KL divergence and maximum likelihood estimation. We highlight the differences between a standard MLE approach and I-Con. In short, although I-Con has a maximum likelihood interpretation, its specific functional form allows it to unify both unsupervised and supervised methods in a way that elucidates the key structures that are important for deriving new representation learning losses. This is in contrast to the commonly known connection between MLE and KL divergence minimization, which does not focus on pairwise connections between datapoints and does not provide as much insight for representation learners. To see this we note that the conventional connection between MLE and KL minimization is as follows:

$$\theta_{\text{MLE}} = \arg \min_{\theta} D_{\text{KL}}(\hat{P}||Q_{\theta}),$$

where the empirical distribution, $\hat{P}$, is defined as:

$$\hat{P}(x) = \frac{1}{N} \sum_{i=1}^{N} \delta(x - x_i),$$

where $\delta(x - x_i)$ is the Dirac delta function. The classical KL minimization fits a parameterized model family to an empirical distribution. In contrast the I-Con equation:

$$\mathcal{L}(\theta, \phi) = \int_{i \in \mathcal{X}} D_{\text{KL}} \left( p_{\theta}(\cdot|i) || q_{\phi}(\cdot|i) \right)$$

Operates on conditional distributions and captures an "average" KL divergence instead of a single KL divergence. Secondly, I-Con explicitly involves a computation over neighboring datapoints which does not appear in the aforementioned equation. This decomposition of methods into their actions on their neighborhoods makes many methods simpler to understand, and makes modifications of these methods easier to transfer between domains. It also makes it possible to apply this theory to unsupervised problems where empirical supervisory data does not exist. Furthermore some methods, like DINO, do not share the exact functional form of I-Con, and suffer from various difficulties like collapse which need to be handled with specific regularizers. This shows that I-Con is not just a catchall reformulation of MLE, but is capturing a specific functional form shared by several popular learners.

## I  ON I-CON'S HYPERPARAMETERS

One important way that I-Con removes hyperparameters from existing works is that it does not rely on things like entropy penalties, activation normalization, activation sharpening, or EMA stabilization to avoid collapse. The loss is self-balancing in this regard as any way that it can improve the learned distribution to better match the target distribution is "fair game". This allows one to generalize certain aspects of existing losses like InfoNCE. In I-Con info NCE looks like fixed-width Gaussian kernels mediating similarity between representation vectors. In I-Con it's trivial to generalize these Gaussians to have adaptive and learned covariances for example. This allows the network to select its own level of certainty in representation learning. If you did this naively, you would need to ensure the loss function doesn't cheat by making everything less certain.

Nevertheless I-Con defines a space of methods depending on the choice of p and q. The choice of these two distributions becomes the main source of hyperparameters we explore. In particular our experiments change the structure of the supervisory signal (often p). For example, in a clustering experiment changing p from "Gaussians with respect to distance" to "graph adjacency" transforms

K-Means into Spectral clustering. It's important to note that K-means has benefits over Spectral clustering in certain circumstances and vice-versa, and there's not necessarily a singular "right" choice for p in every problem. Like many things in ML, the different supervisory distributions provide different inductive biases and should be chosen thoughtfully. We find that this design space makes it easier to build better performing supervisory signals for specific important problems like unsupervised image classification on ImageNet and others.

