# OpenReview forum: "I-Con: A Unifying Framework for Representation Learning"
_ICLR.cc/2025/Conference — ICLR 2025 Poster_

### Official Review · Reviewer_AnKo · 2024-10-26

**Soundness:** 3
**Presentation:** 4
**Contribution:** 3
**Rating:** 8
**Confidence:** 3

**Summary:**

This paper provides a unified framework I-Con that can recover many state-of-the-art approaches in various ML tasks as special examples. The main idea behind the I-Con framework is to minimize the KL divergence between two distributions: a "neighborhood" distribution that is given by data and a proposed distribution that encodes some supervisory information, which is similar to the idea behind variational approaches.

**Strengths:**

1. The paper is very well written and well organized, which conducts a comprehensive survey across different fields and insightfully extracts the similarities behind state-of-the-art methods.
2. It is novel to transfer the proposed distribution across domains and the improvement in performance shown in Section 4 is solid.

**Weaknesses:**

1. The connection with variational Bayesian methods can be elaborated, where the main idea is to find a proper proposed distribution q_{\phi}.
2. This paper provides a comprehensive survey, but it would be better if the authors could have a paragraph to compare with other survey papers, e.g.
	- Tschannen, Michael, Josip Djolonga, Paul K. Rubenstein, Sylvain Gelly, and Mario Lucic. "On mutual information maximization for representation learning." arXiv preprint arXiv:1907.13625 (2019).

**Questions:**

1. [Naming] In terms of I-Con = Information Contrastive learning, I was wondering if it would be more precise to relate the name to "divergence" or "cross-entropy" since the I-Con framework is essentially the KL minimization and nature of "contrastive learning" is not very intuitive without specification of the proposed distribution q_{\phi}.
2. [Experiments] It is interesting to the improvement in performance when transferring the ideas of proposed q_{\phi} across domains. I was wondering if the improvement is also significant beyond the domain of clustering. It would also be good if some related work that transfers the SOTA q_{\phi} across domains could be provided.
3. Please see "weakness" section. I'll be happy to adjust my score if the connection with variation Bayesian methods can be elaborated.

---

> ### Author Response · Authors · 2024-11-25
>
> We thank the reviewer for their thoughtful feedback and have worked to address each of the key concerns raised below. In addition we also added several other key improvements that we detail in the global response. If the reviewer finds these additions helpful we kindly request that they raise their score to help us share this work with the community.
>
> ### Adding experiments beyond the domain of clustering
>
> In a new series of experiments we show that I-Con not only improves clusterers, but can improve contrastive feature learners. As requested by reviewers this experiment also demonstrates that I-Con works on an additional backbone and datasets. We follow the contrastive feature learning experiment put forth by Hu et al. at ICLR 2023. In particular we train a ResNet34 network contrastively on CIFAR10 and evaluate its linear probe and KNN performance on both CIFAR10 and CIFAR100. We include a detailed description of this experimental setting and its results in the Supplement. In summary, we show that our distribution improvement methods like debiasing and replacing gaussian kernels with cauchy kernels dramatically improves feature quality both in terms of linear probe accuracy and KNN accuracy.
>
>
> | **Method**                          | **CIFAR10 (in distribution)** |                        | **CIFAR100 (out of distribution)** |                     |
> |-------------------------------------|-------------------------------|------------------------|-------------------------------------|---------------------|
> |                                     | Linear Probing               | KNN                    | Linear Probing                      | KNN                 |
> | **$q_\phi$ is a Gaussian Distribution**                                                                                                 |
> | SimCLR [Chen et al., 2020]          | 77.79                        | 80.02                  | 31.82                               | 40.27               |
> | DCL [Chuang et al., 2020]           | 78.32                        | 83.11                  | 32.44                               | 42.10               |
> | **Our Debiasing $\alpha = 0.2$**    | 79.50                        | 84.07                  | 32.53                               | 43.19               |
> | **Our Debiasing $\alpha = 0.4$**    | 79.07                        | 85.06                  | 32.53                               | **43.29**           |
> | **Our Debiasing $\alpha = 0.6$**    | 79.32                        | 85.90                  | 30.67                               | 29.79               |
> | **$q_\phi$ is a Student's t-distribution**                                                                                             |
> | t-SimCLR [Hu et al., 2022]          | 90.97                        | 88.14                  | 38.96                               | 30.75               |
> | DCL [Chuang et al., 2020]           | Diverges                     | Diverges               | Diverges                            | Diverges            |
> | **Our Debiasing $\alpha = 0.2$**    | 91.31                        | 88.34                  | 41.62                               | 32.88               |
> | **Our Debiasing $\alpha = 0.4$**    | 92.70                        | 88.50                  | **41.98**                           | 34.26               |
> | **Our Debiasing $\alpha = 0.6$**    | **92.86**                    | **88.92**              | 38.92                               | 32.51               |
>
>
> ### Comparing I-Con to Related Survey Papers, Such as Tschannen et al.
>
> In the revised version of our paper, we included a short discusses on Tschannen et al.'s paper. Their work critically examines the role of mutual information (MI) maximization in representation learning, highlighting that the success of MI-based methods often arises from architectural choices and parameterization of MI estimators rather than from MI maximization itself.
>
> Our I-Con framework differs from Tschannen et al.'s focus in several key ways:
>
> 1. **Scope and Unification**:
>    While Tschannen et al. critique the effectiveness of MI maximization, our framework provides a unifying equation that encompasses a wide range of methods across clustering, spectral methods, dimensionality reduction, contrastive learning, and supervised learning.
>
> 2. **Avoiding MI Estimation Challenges**:
>    We sidestep the difficulties associated with estimating MI in high-dimensional spaces by formulating our objective as minimizing the KL divergence between conditional distributions. This approach avoids the estimation issues highlighted by Tschannen et al.
>
> By including this comparison, we aim to highlight the contributions of our work in advancing the understanding and practice of representation learning beyond the limitations identified in prior studies.

---

> > ### Author Response · Authors · 2024-11-25
> >
> > ### Expanded Elaboration on the Connection to Variational Bayesian Methods
> >
> > Thank you for highlighting the potential connection between our I-Con framework and variational Bayesian (VB) methods. We agree that both approaches involve optimizing over a family of distributions $q_\phi$ to approximate a target distribution, and we appreciate the opportunity to elaborate on this connection.
> >
> > In variational Bayesian methods, the goal is to approximate an intractable posterior distribution $ p(z \mid x) $ by introducing a variational distribution $q_\phi(z)$ and minimizing the KL divergence $\text{KL}(q_\phi(z) \parallel p(z \mid x))$. This optimization allows for tractable inference in complex probabilistic models by finding the best approximation within a chosen family of distributions $q_\phi$.
> >
> > Similarly, our I-Con framework seeks to maximize mutual information by introducing a variational distribution $q_\phi(j \mid i)$ (the learned distribution) and aligning it with a supervisory distribution $p_\theta(j \mid i)$ by minimizing the KL divergence between them:
> >
> > $$
> > \mathcal{L}(\theta, \phi) = \int_{i \in X} \text{KL}(p_\theta(\cdot \mid i) \parallel q_\phi(\cdot \mid i))
> > $$
> >
> > In both cases, the optimization involves selecting a suitable \( q_\phi \) to approximate or align with a target distribution. However, there are key differences:
> >
> > 1. **Nature of Target Distributions**:
> >    In VB methods, the target distribution $p(z \mid x)$ is typically the true posterior, which is often intractable. In our framework, $p_\theta(j \mid i)$ is a supervisory distribution that can be constructed based on data relationships (e.g., similarities, neighborhoods) and is usually tractable.
> >
> > 2. **Purpose and Application**:
> >    VB methods focus on probabilistic inference and learning in latent variable models, aiming to approximate posterior distributions. In contrast, I-Con is designed for representation learning by aligning conditional distributions over observed data, unifying various methods across different domains.
> >
> > 3. **Role of KL Divergence**:
> >    While both frameworks use KL divergence as a measure of dissimilarity between distributions, in VB methods, the KL divergence is often reversed (i.e., $\text{KL}(q_\phi(z) \parallel p(z \mid x))$) compared to our use in I-Con (i.e., $ \text{KL}(p_\theta(\cdot \mid i) \parallel q_\phi(\cdot \mid i)) $).
> >
> > To clarify this connection, we will include a dedicated paragraph in our paper that elaborates on the parallels and distinctions between the I-Con framework and variational Bayesian methods. This addition will provide readers with a deeper understanding of the theoretical underpinnings of our approach and how it relates to existing probabilistic frameworks.
> >
> >
> > ### Suggested Using a Name That “Better Reflects Its Basis in 'Divergence' or 'Cross-Entropy,' as 'Contrastive Learning' in the Current Name Is Less Intuitive.”
> >
> > Thanks for this suggestion! Though I-Con is our preliminary name, we will continue searching for a simple and easy-to-remember name that better captures the semantics of the idea for the camera-ready version. We’ll keep these suggestions in mind as we workshop possible new names.

---

> > > ### Comment · Reviewer_AnKo · 2024-11-26
> > > **Reply to Authors**
> > >
> > > Thanks a lot for the detailed reply! I believe the authors will have a comprehensive comparison with variational methods in an updated draft. Also, the added simulation results extend beyond the regime of clustering. I'll raise my score.

---

> > > > ### Author Response · Authors · 2024-11-28
> > > > **Thank you!**
> > > >
> > > > Thank you for your thoughtful comments and for pushing us to investigate the interesting connections with VB. We greatly appreciate you raising your score.

---

### Official Review · Reviewer_FvTp · 2024-10-29

**Soundness:** 3
**Presentation:** 2
**Contribution:** 2
**Rating:** 6
**Confidence:** 3

**Summary:**

This paper addresses the challenge of the variety of unsupervised representation learning methods by introducing I-Con, a unified interpretation grounded in a single mathematical framework. The authors provide a theoretical proof that I-Con unifies existing methods by minimizing the KL divergence between the supervisory and parameterized neighborhood distribution. Furthermore, the authors effectively transferred perspectives of various representation learning approaches into a state-of-the-art unsupervised classification network and showed both quantitative and qualitative improvements.

**Strengths:**

- This paper discusses various methods in representation learning in terms for their proposal.
- The proposed framework is well demonstrated throughout the paper including theoretical evidences.
- The proposed method showed significant improvements compared to state-of-the-art unsupervised classification network.

**Weaknesses:**

- The motivation for the need for a unified framework of (unsupervised) representation learning is not clear. As discussed in introduction section, “difficult to understand which particular loss function to choose for a given problem or domain”, does not directly match with the proposed method, as I-CON still contains the choice of $p_{\theta}$ for each existing method and does not provide a suitable loss function for given domain.
- The term ‘I-Con’ is ambiguous, as it is used with two different meanings. In section 3.1, term ‘I-Con’ represents a unified representation learning algorithm that can that can serve as the foundation for various methods. However in section 3.2, it is also used to describe integrated (new) representation learning method, causing confusion in the following results. To avoid confusion, use distinct terms to clearly differentiate between I-Con as a framework and I-Con as a specific method.
- Section 3.3 is not easy to follow, especially the sub-section ‘Adaptive neighborhoods’. Paraphrasing the explanations and providing a more detailed demonstration would enhance clarity for the readers.
- Random seeds are not considered in the experimental results. Providing results of multiple runs (e.g., 5 or 10) with standard variations (or error bars) would give stronger evidence for the claimed significant improvements.

**Questions:**

### Questions
- Theoretical evidences for I-Con across various method are given (appendix). However, are there any experimental results in practical environments (e.g., InfoNCE on ImageNet) that shows the I-CON framework can reproduce (or improve) the original representation learning methods?
- In Figure 3, is there a reason for using different dataset (ImageNet-1K and subset of MNIST)? Why not use ImageNet for both the quantitative and qualitative results (or vice versa)?
- Is I-Con (Ours) stated in Table 2 result of integrated method discussed in Section 3.3, or is it the best among them? Table 3 shows results of each transferred intuitions, and results of I-Con (Ours) stated in Table 2 appears to be the best among them. Referring to the best among them as I-Con (Ours) in Table 2 may not be a fair comparison.
- As stated in Section 4.2, I-Con has claimed to have the benefit of not requiring additional hyper-parameter tuning compared to TEMI and SCAN. However, doesn’t I-Con require hyper-parameter tuning, such as /alpha and n-walks (Figure 4, Table 4), for notable improvements?
- DiNO[1] also showed results of k-NN classifiers[2] in the paper. Could results in Table 3 maintain its gains when using k-NN classifiers[2] as stated in [1]?
### Modifications
- Table 2 in the first line in Section 4.2, do not have a link to it.
- Like in t-SNE this addition helps… → Like in t-SNE, this addition helps…

[1] Mathilde Caron, Hugo Touvron, Ishan Misra, Herv´ e J´ egou, Julien Mairal, Piotr Bojanowski, and Armand Joulin. Emerging properties in self-supervised vision transformers, 2021.

[2] Zhirong Wu, Yuanjun Xiong, Stella X Yu, and Dahua Lin. Unsupervised feature learning via non-parametric instance discrimination. In CVPR, 2018.

---

> ### Author Response · Authors · 2024-11-25
>
> We thank the reviewer for their thoughtful feedback and have worked to address each of the key concerns raised below. In addition we also added several other key improvements that we detail in the global response. If the reviewer finds these additions helpful we kindly request that they raise their score to help us share this work with the community.
>
> ### Correct naming of methods in main table and using a single version of the method
>
> This is a good point and we have adjusted the naming in the table to InfoNCE Clustering to make it clear that this is a derived clustering method from the broader I-Con framework. We also now use the same version of the method (with EMA, Walks, and Debiasing) across the entries of Table 2 as requested.
>
> ### Report multiple seeds and error bars for clustering results to strengthen empirical evidence.
>
> We have added error bars corresponding to the standard error of the mean in table 2. We calculate these error bars from 5 different seeds. The main conclusions of the work still hold, and we maintained the SOTA across all seeds.
> Evaluate whether the I-Con framework can reproduce or improve original methods like InfoNCE on ImageNet.
> Yes, not only is InfoNCE a special case of the I-Con framework, it can also be improved by I-Con. In particular, similar to how T-SNE improves SNE by using cauchy distributions, we see a similar improvement in contrastive learning. Secondly, we find that our debiasing operator also improves classic vanilla InfoNCE. We add a variety quantitative experiments verifying these claims in the supplement.
>
> ### On replacing MNIST qualitative Figures with Imagenet
>
> We chose MNIST because of the small number of classes which made it easy to visualize, we have since refined our visualization to apply to imagenet and have updated the figure accordingly.

---

> > ### Author Response · Authors · 2024-11-25
> >
> > ### Clarifying the role of hyper-parameter tuning
> >
> > We added a section to the appendix clarifying the role of hyper-parameters in I-Con. One important way that I-Con removes hyperparameters from existing works is that it does not rely on things like entropy penalties, activation normalization, activation sharpening, or EMA stabilization to avoid collapse. The loss is self-balancing in this regard as any way that it can improve the learned distribution to better match the target distribution is “fair game”. This allows one to generalize certain aspects of existing losses like InfoNCE. In I-Con info NCE looks like fixed-width gaussian kernels mediating similarity between representation vectors. In I-Con it’s trivial to generalize these gaussians to have adaptive and learned covariances for example. This allows the network to select its own level of certainty in representation learning. If you did this naively, you would need to ensure the loss function doesn't cheat by making everything less certain.
> >
> > Nevertheless I-Con defines a space of methods depending on the choice of p and q. The choice of these two distributions becomes the main source of hyperparameters we explore. In particular our experiments change the structure of the supervisory signal (often p). For example, in a clustering experiment  changing p from “gaussians with respect to distance” to “graph adjacency” transforms K-Means into Spectral clustering. It’s important to note that K-means has benefits over Spectral clustering in certain circumstances and vice-versa, and there’s not necessarily a singular “right” choice for p in every problem. Like many things in ML, the different supervisory distributions provide different inductive biases and should be chosen thoughtfully. We find that this design space makes it easier to build better performing supervisory signals for specific important problems like unsupervised image classification on ImageNet and others.
> >
> > ### Clarify Section 3.3
> > We have clarified section 3.3 and hopefully made the key intuitions clearer.

---

> > > ### Comment · Reviewer_FvTp · 2024-11-26
> > >
> > > Thank you for responding to my review. I appreciate the authors' efforts to address my concerns.
> > >
> > > However, several parts remaining unclear to me:
> > >
> > > > Clarifying the role of hyper-parameter tuning
> > > >
> > >
> > > I agree that compared to previous works, I-Con can self-balance the certainty in representation learning without additional methods (e.g., entropy penalties, activation normalization, activation sharpening, or EMA stabilization). However, I still think this conflicts with your claims in ablation study at Section 4.3, which highlights the sensitivity to $\alpha$ and propagation size—hyper-parameters that still require tuning. Is it possible to claim that I-Con doesn’t require additional regularizations rather than hyper-parameter tuning?
> > >
> > > > Report multiple seeds and error bars for clustering results to strengthen empirical evidence.
> > > >
> > >
> > > In Table 2, why not report error bars for baselines and scores in Table 3?
> > >
> > > \
> > > For the following questions, I can’t find the response to my questions. Please let me know if I have misunderstood something here.
> > >
> > > > Is I-Con (Ours) stated in Table 2 result of integrated method discussed in Section 3.3, or is it the best among them? Table 3 shows results of each transferred intuitions, and results of I-Con (Ours) stated in Table 2 appears to be the best among them. Referring to the best among them as I-Con (Ours) in Table 2 may not be a fair comparison.
> > > >
> > >
> > > > DiNO[1] also showed results of k-NN classifiers[2] in the paper. Could results in Table 3 maintain its gains when using k-NN classifiers[2] as stated in [1]?
> > > >
> > >
> > > > Table 2 in the first line in Section 4.2, do not have a link to it.
> > > >
> > >
> > > \
> > > I would like to once again thank the authors for their responses to my review and encourage them to resolve the remaining discussions successfully during the rebuttal period.

---

> ### Author Response · Authors · 2024-11-28
>
> Thank you for your review and thoughtful comments. We have made several updates to clarify and improve the paper.
>
> ---
>
> > I agree that compared to previous works, I-Con can self-balance the certainty in representation learning without additional methods (e.g., entropy penalties, activation normalization, activation sharpening, or EMA stabilization). However, I still think this conflicts with your claims in the ablation study in Section 4.3, which highlights the sensitivity to propagation size and other hyper-parameters that still require tuning. Is it possible to claim that I-Con doesn’t require additional regularizations rather than hyper-parameter tuning?
>
> Yes, not requiring regularizers is a correct way of saying it, we updated the text in that section 4.2 to better reflect that view and make it less confusing!
>
> ---
>
> > In Table 2, why not report error bars for baselines and scores in Table 3?
>
> Due to time constraints during the initial submission, we prioritized reporting the most significant row. However, in this revision, we have updated all entries in Table 3 to include error bars and have added error bars to Figure 4 as well. Thank you for your patience as we worked on this!
>
> ---
>
> > Is I-Con (Ours) stated in Table 2 the result of the integrated method discussed in Section 3.3, or is it the best among them? Table 3 shows results of each transferred intuition, and results of I-Con (Ours) stated in Table 2 appear to be the best among them. Referring to the best among them as I-Con (Ours) in Table 2 may not be a fair comparison.
>
> Thank you for raising this point. We have updated the numbers in Table 2 to clarify that "(Ours)" refers to the model with EMA, walks of length $2$, and $\alpha = 0.4$.
>
> ---
>
> > DiNO [1] also showed results of k-NN classifiers [2] in the paper. Could results in Table 3 maintain its gains when using k-NN classifiers as stated in [1]?
>
> Table 3 presents the Hungarian clustering accuracy, the standard method for evaluating unsupervised classification systems. Since we are not learning a continuous vector specifically designed for retrieval in these experiments, we believe that such a metric would not be meaningful. However, in our new feature learning experiments on CIFAR-10, CIFAR-100, STL-10, and Oxford-IIIT Pet (the last two were added in this revision), we have included k-NN accuracy results for both in-distribution and out-of-distribution evaluations.
>
> ---
>
> > Table 2 in the first line in Section 4.2 does not have a link to it.
>
> Thank you this got lost in the rebuttal whirlwind, we have since added it!
>
> ---
>
> Thanks for your lovely round of feedback and clarifications! We hope this next revision clears up the remaining points.

---

> > ### Comment · Reviewer_FvTp · 2024-11-29
> >
> > Thank you to the authors for their efforts in addressing my concerns. Based on their responses, I am raising my score from 5 to 6. I hope the authors can incorporate these additional points into the final version of the paper.

---

> > > ### Author Response · Authors · 2024-11-29
> > > **Thank you!**
> > >
> > > Thank you for your valuable feedback and for raising your score. We appreciate your suggestions and will ensure they are reflected in the paper.

---

### Official Review · Reviewer_ioHt · 2024-10-31

**Soundness:** 2
**Presentation:** 3
**Contribution:** 3
**Rating:** 5
**Confidence:** 3

**Summary:**

The manuscript attempts to build a universal framework for loss functions corresponding to different representation learning methods. It minimizes an integrated KL divergence between two marginal distributions: the supervisory and learned representations. The proposed framework encompass 11 machine learning methods, including dimensionality reduction, contrastive learning, clustering, and supervised learning, among others.

**Strengths:**

One advantage is that it can leverage losses from other domains to enhance a particular domain method within this framework. For instance, in experiments, spectral clustering, t-SNE, debiased contrastive learning, and SCAN were utilized to develop an unsupervised image classification system.

**Weaknesses:**

- It falls short of truly encompassing all representation learning methods, such as autoencoder-based representation learning, masked pretraining-based representation learning, and other general self-supervised representation learning methods. Therefore, the proposed framework does not entirely unify all representation learning methods. I believe that this unified framework is for clustering methods but not a generic representation learning framework.
- The motivation/benefits for unifying representation learning methods claimed in this paper do not entirely convince me personally. Could you give more discussions on whether this unified framework provides advantages beyond facilitating cross-domain insights? For example, does it provide a new method to address all limitations of different representation learning frameworks once?
- Typically, evaluating representation learning methods involves assessing if a model has learned good representations that generalize well across various downstream tasks. However, I did not observe such experiments.
- The choice of data augmentation strategies is crucial in contrastive learning methods. Are data augmentation-based contrastive learning methods also unified within this framework?
- Are there differences in the task purposes between clustering and contrastive learning tasks? My understanding is that clustering aims to discover different clusters, while contrastive learning aims to learn generic representations that can be easily transferred to different downstream tasks. Could you provide further discussion on this?

Additional minor suggestions:
- Figure 1 is not easily read. The cluster in the right part should be labeled as "D)".

**Questions:**

Please see Weakness.

---

> ### Author Response · Authors · 2024-11-25
>
> We thank the reviewer for their thoughtful feedback and have worked to address each of the key concerns raised below. In addition we also added several other key improvements that we detail in the global response. If the reviewer finds these additions helpful we kindly request that they raise their score to help us share this work with the community.
>
> ## 1- New Series of Experiments Demonstrating that I-Con improves Feature Learning and works on different backbones and datasets.
>
> In a new series of experiments we show that I-Con not only improves clusterers, but can improve contrastive feature learners. As requested by reviewers this experiment also demonstrates that I-Con works on an additional backbone and datasets. We follow the contrastive feature learning experiment put forth by Hu et al. at ICLR 2023. In particular we train a ResNet34 network contrastively on CIFAR10 and evaluate its linear probe and KNN performance on both CIFAR10 and CIFAR100. We include a detailed description of this experimental setting and its results in the Supplement. In summary, we show that our distribution improvement methods like debiasing and replacing gaussian kernels with cauchy kernels dramatically improves feature quality both in terms of linear probe accuracy and KNN accuracy.
>
> | **Method**                          | **CIFAR10 (in distribution)** |                        | **CIFAR100 (out of distribution)** |                     |
> |-------------------------------------|-------------------------------|------------------------|-------------------------------------|---------------------|
> |                                     | Linear Probing               | KNN                    | Linear Probing                      | KNN                 |
> | **$q_\phi$ is a Gaussian Distribution**                                                                                                 |
> | SimCLR [Chen et al., 2020]          | 77.79                        | 80.02                  | 31.82                               | 40.27               |
> | DCL [Chuang et al., 2020]           | 78.32                        | 83.11                  | 32.44                               | 42.10               |
> | **Our Debiasing $\alpha = 0.2$**    | 79.50                        | 84.07                  | 32.53                               | 43.19               |
> | **Our Debiasing $\alpha = 0.4$**    | 79.07                        | 85.06                  | 32.53                               | **43.29**           |
> | **Our Debiasing $\alpha = 0.6$**    | 79.32                        | 85.90                  | 30.67                               | 29.79               |
> |-------------------------------------|-------------------------------|------------------------|-------------------------------------|---------------------|
> | **$q_\phi$ is a Student's t-distribution**                                                                                             |
> | t-SimCLR [Hu et al., 2022]          | 90.97                        | 88.14                  | 38.96                               | 30.75               |
> | DCL [Chuang et al., 2020]           | Diverges                     | Diverges               | Diverges                            | Diverges            |
> | **Our Debiasing $\alpha = 0.2$**    | 91.31                        | 88.34                  | 41.62                               | 32.88               |
> | **Our Debiasing $\alpha = 0.4$**    | 92.70                        | 88.50                  | **41.98**                           | 34.26               |
> | **Our Debiasing $\alpha = 0.6$**    | **92.86**                    | **88.92**              | 38.92                               | 32.51               |
>
>
> ## 2- Proving that MDS, PCA (Linear Autoencoders), Masked Language Modeling, Spectral Clustering, Mocov3, and Triplet Loss are unified by I-Con
>
> We introduce new theorems that unify 6 additional methods with the I-Con framework. In particular we show that MDS and SNE can be unified in the limit of large gaussian scales, which is a novel theoretical result first proved in this work. As requested by reviewer iHot, this result allows us to integrate linear autoencoders (PCA) into I-Con.  As requested by reviewer iHot we also show masked language modeling can be considered in the I-Con framework. Intuitively this  is a simple corollary of our existing proof connecting Cross Entropy with I-Con, except replacing class embeddings with embeddings for masked tokens. Finally, we show that the commonly used triplet loss for contrastive learning emerges as in the limit of small width gaussians in I-Cons InfoNCE connection. These additional proofs can be found in the supplement.

---

> ### Author Response · Authors · 2024-11-25
>
> ### Elaborating on the difference between representation learning, clustering, and contrastive learning
> We want to take the opportunity to clear up some questions and misunderstandings about I-Con and representation learning more broadly. In particular
> 1. I-Con does already unify several representation learning methods. And is not just a unification of clustering methods. In particular I-Con unifies many methods in Dimensionality Reduction (t-SNE, SNE, In this revision MDS, PCA (Linear AutoEncoding), Contrastive Learning (SimCLR, MocoV3), and in this revision we added masked pretraining as well.
> 2. We don’t claim to unify every representation learning method, but we do show a diverse collection of popular algorithms are unified by I-Con. For example DinoV2 is not an I-Con method and suffers from collapse that needs to be rectified through activation sharpening.
> 3. When we say “representation learning” we usually refer to the broader context where one can learn any kind of representation, discrete, continuous, low-dim etc. The difference between contrastive feature learning and contrastive cluster learning is whether the method learns vectors in R^n or cluster assignments. These two methods need to be evaluated differently.
> 4. In our original submission we did not use linear probe evaluation strategies because they apply to continuous representations.
> 5. For our clustering experiments we use the commonly accepted hungarian accuracy.
> Nevertheless we added additional experiments with linear probe and KNN analysis to show I-Con can improve contrastive feature learners ad well as clusterers.
>
> ### Clarifying that data augmentation-based contrastive learning methods are already unified within the I-Con framework.
>
> I-Con already unifies data-augmentation based contrastive methods and we point the reviewer to our analysis of the InfoNCE loss in the appendix.  We also tried to make these connections a bit more obvious in the main text.  To briefly recap here, augmentations are a particular type of supervisory distribution, p, where p(i|j) = 1 iff i and j are different augmentations of the same image or object. In this setting I-Con reproduces standard InfoNCE style learning over augmentation pairs popularized in SimCLR.
>
> ### Improving motivation of Unifying Framework
> We add a section further motivating the I-Con framework and detailing some of the things a unified framework provides. As noted by reviewer xJUE, this unification “not only provides a deeper understanding of these methods but also opens up the possibility of creating new methods by mixing and matching components.” We explicitly used this property to discover new improvements to both clustering and representation learners.  In short, I-Con acts like a periodic table of machine learning losses. With this periodic table we can more clearly see the implicit assumptions of each method by breaking down modern ML losses into more simple components: pairwise conditional distributions $p$ and $q$.
>
> One particular example of how this opens new possibilities is with our generalized debiasing operation. THrough our experiments we show adding a slight constant linkage between datapoints improves both stability and performance across clustering and feature learning. Unlike prior art, which only applies to specific feature learners, our debiasers can improve clusterers, feature learners, spectral graph methods, and dimensionality reducers.
>
> FInally it allows us to discover novel theoretical connections by compositionally exploring the space, and considering limiting conditions. We use I-Con to help derive a novel theoretical equivalences between K-Means and constrastive learning (initial work) and between MDS, PCA, and SNE (In this rebuttal and in the new supplement). Reviewer xJUE mentioned that transferring ideas between methods is standard in research, but in our view it becomes much simpler to do this if you know methods are equivalent. Previously, it might not be clear how exactly to translate an insight like changing gaussian distributions to Cauchy distributions in the upgrade from SNE to T-SNE has any effect on clustering or representation learning. In I-Con it becomes clear to see that similarly softening clustering and representation learning distributions can improve performance and debias representations.
>
> ### Fixing Figure 1
> Thank you for raising this issue, we have corrected the label in Figure 1. If you would like to see the main figure modified in any other way please let us know.

---

> > ### Author Response · Authors · 2024-11-28
> >
> > ## Additional Experiments: Demonstrating I-Con's Effectiveness Across Backbones and Datasets
> > We are happy to share results from additional experiments conducted to evaluate I-Con's performance on feature learning. In addition to our new feature learning experiments on CIFAR-10 and CIFAR-100, we included evaluations on STL-10 (in-domain) and Oxford-IIT Pets (out-of-domain) to support the generalization of our framework beyond clustering. Detailed descriptions of the experimental setup, along with result analyses and visualizations, are provided in the Supplement.
> >
> > | **Method**                        | **STL-10 (In-Distribution)** |                         | **Oxford-IIIT Pets (Out-of-Distribution)** |                         |
> > |-----------------------------------|-----------------------------|-------------------------|--------------------------------------------|-------------------------|
> > |                                   | Linear Probing              | KNN                     | Linear Probing                             | KNN                     |
> > | **$q_\phi$ is a Gaussian Distribution**                                                                                                                           |
> > | SimCLR [Chen et al., 2020]        | 77.71                       | 74.92                   | 74.80                                      | 71.48                   |
> > | DCL [Chuang et al., 2020]         | 78.42                       | 75.03                   | 74.41                                      | 70.22                   |
> > |                                   |                             |                         |                                            |                         |
> > | **$q_\phi$ is a Student's t-Distribution**                                                                                                                        |
> > | t-SimCLR [Hu et al., 2022]        | 85.11                       | 83.05                   | 83.49                                      | 81.41                   |
> > | DCL [Chuang et al., 2020]         | Diverges                    | Diverges                | Diverges                                   | Diverges                |
> > | **Our Debiasing $\alpha = 0.2$**  | 85.94                       | 83.15                   | 84.11                                      | 81.15                   |
> > | **Our Debiasing $\alpha = 0.4$**  | 86.13                       | **84.14**               | 84.07                                      | **84.13**               |
> > | **Our Debiasing $\alpha = 0.6$**  | **87.18**                   | 83.58                   | **84.51**                                  | 83.04                   |
> >
> > In summary, the results support that methods focused on distribution improvements—such as debiasing and replacing Gaussian kernels with Cauchy kernels—significantly enhance feature quality across datasets evident both in linear probe accuracy and KNN accuracy.

---

> > > ### Author Response · Authors · 2024-11-30
> > >
> > > Thank you for the feedback that improved our paper. We have added new sections and results for the points you have raised. If these additional analyses and results have improved the work we kindly ask that you raise your score to an accept. Otherwise, please let us know if there is anything else you would like us to add. Thank you very much for your time and consideration!

---

> > > > ### Author Response · Authors · 2024-12-01
> > > >
> > > > Thank you for taking the time to review our work, as we approach the last day of the discussion period we wanted to follow up to see if it was possible to improve our score if you found our additional analyses, clarifications, and descriptions helpful. Otherwise please let us know if theres anythign else that can be done to address concerns. Thank you!

---

### Official Review · Reviewer_xJUE · 2024-11-01

**Soundness:** 4
**Presentation:** 4
**Contribution:** 4
**Rating:** 6
**Confidence:** 4

**Summary:**

The paper presents a novel framework called I-Con, which unifies various machine learning methods, including clustering, contrastive learning, dimensionality reduction, and supervised learning, under a single equation based on the Kullback-Leibler (KL) divergence between two distributions: the supervisory and learned representations. The authors demonstrate that many existing methods are instances of this unified framework, providing a mathematical foundation that enables the transfer of ideas across domains. The paper also presents new state-of-the-art results in unsupervised image classification on the ImageNet-1K dataset, achieving an 8% improvement over previous methods. Multiple proofs and empirical evaluations back these claims, illustrating the broad applicability of I-Con to various learning problems.

**Strengths:**

- Theoretical Contributions: The I-Con framework offers a unified theoretical foundation for a wide range of machine learning methods, showing that several existing approaches can be viewed as specific instances of a general KL divergence minimization problem. This not only provides a deeper understanding of these methods but also opens up the possibility of creating new methods by mixing and matching components.
- Comprehensive Theoretical Justification: The paper provides detailed mathematical proofs, connecting 11 different approaches to the I-Con framework.
- Cross-Domain Applicability: By unifying disparate learning paradigms (clustering, dimensionality reduction, contrastive learning, etc.), the framework facilitates the transfer of techniques across domains, allowing innovations from one area (e.g., contrastive learning) to improve methods in another (e.g., clustering).
- Empirical Performance: The framework's potential is validated through unsupervised image classification on ImageNet-1K, where the I-Con-based method outperforms state-of-the-art methods, demonstrating its practical significance.

**Weaknesses:**

- Innovation and Breadth: While the authors introduce the I-Con framework, which attempts to unify multiple existing learning methods, the actual novelty and breadth of this framework might be questionable. Many of the connections between the methods discussed have already been noted in previous works, and the core contribution seems to be more about integrating existing ideas rather than offering fundamentally new insights. Actually, most methods and loss functions can be implemented in a likelihood framework, which is equivalent to minimizing KL distance.
- Limited Guidance According to The Theorem: Several techniques like adaptive neighborhoods, debiased contrastive learning, neighbor propagation are transfered to create a state-of-the-art unsupervised image classification system. However, these methods can be thought out without the I-Con framework as many articles have established. Actually, inspiring or taking ideas from other domains is natural during research. Thus, based on the theorem, it will be more valuable to discuss how to choose the mechanism according to the I-Con framework.
- Limited Generalization and Insufficient Experimental Validation: While the paper provides some experimental results demonstrating improvements in unsupervised classification, these are mainly focused on clustering on ImageNet-1K. The generalizability of the I-Con framework to other applications and datasets is still underexplored, lacking broader experimental comparisons across a variety of tasks and datasets.
I would be glad to improve my score if my concerns could be addressed.

**Questions:**

See weakness.

---

> ### Author Response · Authors · 2024-11-25
>
> We thank the reviewer for their thoughtful feedback and have worked to address each of the key concerns raised below. In addition we also added several other key improvements that we detail in the global response. If the reviewer finds these additions helpful we kindly request that they raise their score to help us share this work with the community.
>
> ### New Series of Experiments Demonstrating that I-Con improves Feature Learning and works on different backbones and datasets.
>
> In a new series of experiments we show that I-Con not only improves clusterers, but can improve contrastive feature learners. As requested by reviewers this experiment also demonstrates that I-Con works on an additional backbone and datasets. We follow the contrastive feature learning experiment put forth by Hu et al. at ICLR 2023. In particular we train a ResNet34 network contrastively on CIFAR10 and evaluate its linear probe and KNN performance on both CIFAR10 and CIFAR100. We include a detailed description of this experimental setting and its results in the Supplement. In summary, we show that our distribution improvement methods like debiasing and replacing gaussian kernels with cauchy kernels dramatically improves feature quality both in terms of linear probe accuracy and KNN accuracy. [update: we added more datasets (STL-10 and Oxford-IIT Pets) in the latest comment as well!]
>
> | **Method**                          | **CIFAR10 (in distribution)** |                        | **CIFAR100 (out of distribution)** |                     |
> |-------------------------------------|-------------------------------|------------------------|-------------------------------------|---------------------|
> |                                     | Linear Probing               | KNN                    | Linear Probing                      | KNN                 |
> | **$q_\phi$ is a Gaussian Distribution**                                                                                                 |
> | SimCLR [Chen et al., 2020]          | 77.79                        | 80.02                  | 31.82                               | 40.27               |
> | DCL [Chuang et al., 2020]           | 78.32                        | 83.11                  | 32.44                               | 42.10               |
> | **Our Debiasing $\alpha = 0.2$**    | 79.50                        | 84.07                  | 32.53                               | 43.19               |
> | **Our Debiasing $\alpha = 0.4$**    | 79.07                        | 85.06                  | 32.53                               | **43.29**           |
> | **Our Debiasing $\alpha = 0.6$**    | 79.32                        | 85.90                  | 30.67                               | 29.79               |
> |-------------------------------------|-------------------------------|------------------------|-------------------------------------|---------------------|
> | **$q_\phi$ is a Student's t-distribution**                                                                                             |
> | t-SimCLR [Hu et al., 2022]          | 90.97                        | 88.14                  | 38.96                               | 30.75               |
> | DCL [Chuang et al., 2020]           | Diverges                     | Diverges               | Diverges                            | Diverges            |
> | **Our Debiasing $\alpha = 0.2$**    | 91.31                        | 88.34                  | 41.62                               | 32.88               |
> | **Our Debiasing $\alpha = 0.4$**    | 92.70                        | 88.50                  | **41.98**                           | 34.26               |
> | **Our Debiasing $\alpha = 0.6$**    | **92.86**                    | **88.92**              | 38.92                               | 32.51               |
>
>
> ### Proving that MDS, PCA (Linear Autoencoders), Masked Language Modeling, Spectral Clustering, and Triplet Loss are unified by I-Con
>
> We introduce new theorems that unify 6 additional methods with the I-Con framework. In particular we show that MDS and SNE can be unified in the limit of large gaussian scales, which is a novel theoretical result first proved in this work. As requested by reviewer iHot, this result allows us to integrate linear autoencoders (PCA) into I-Con.  As requested by reviewer iHot we also show masked language modeling can be considered in the I-Con framework. Intuitively this  is a simple corollary of our existing proof connecting Cross Entropy with I-Con, except replacing class embeddings with embeddings for masked tokens. We also show that Spectral Clustering emerges naturally from I-Con. Finally, we show that the commonly used triplet loss for contrastive learning emerges as in the limit of small width gaussians in I-Cons InfoNCE connection. These additional proofs can be found in the supplement.

---

> > ### Author Response · Authors · 2024-11-25
> >
> > ### Comparison to KL Minimization in the Maximum Likelihood Context
> > We added a section in our supplement that elaborates more on the connection between MLE, KL Divergence, and I-Con. In short, although I-Con has a maximum likelihood interpretation, its specific functional form allows it to unify both unsupervised and supervised methods in a way that elucidates the key structures that are important for deriving new representation learning losses. This is in contrast to the commonly known connection between MLE and KL divergence minimization, which does not focus on pairwise connections between datapoints and does not provide as much insight for representation learners. To see this we note that the conventional connection between MLE and KL minimization is as follows:
> >
> >
> >
> > $$ \theta_{\text{MLE}} = \arg\min_\theta D_{\text{KL}}(\hat{P} || Q_\theta), $$
> >
> > where the empirical distribution,  $ \hat{P} $ ,  is defined as: \[ \hat{P}(x) = \frac{1}{N} \sum_{i=1}^N \delta(x - x_i), \] where \(\delta(x - x_i)\) is the Dirac delta function. The classical KL minimization fits a parameterized model family to an empirical distribution. In contrast the I-Con equation:
> >
> > $$
> >     \mathcal{L}(\theta, \phi) = \int_{i \in \mathcal{X}} D_{\text{KL}} \left( p_{\theta}(\cdot|i) || q_{\phi}(\cdot|i ) \right)
> > $$
> >
> >
> > Operates on conditional distributions and captures an “average” KL divergence instead of a single KL divergence. Secondly, I-Con explicitly involves a computation over neighboring datapoints which does not appear in the aforementioned equation. This decomposition of methods into their actions on their neighborhoods makes many methods simpler to understand, and makes modifications of these methods easier to transfer between domains. It also makes it possible to apply this theory to unsupervised problems where empirical supervisory data does not exist. Furthermore, some methods, like DINO, do not share the exact functional form of I-Con, and suffer from various difficulties like collapse which need to be handled with specific regularizers. This shows that I-Con is not just a catchall reformulation of MLE, but is capturing a specific functional form shared by several popular learners.
> >
> > ### On novelty
> > We politely rebut the statement that our novelty is “questionable”. Other reviewers (“AnKo”) praised the work’s novelty citing “It is novel to transfer the proposed distribution across domains”. We are the first work to recognize the core functional form underlying 15+ existing loss functions. Several of our connections are not documented in the literature including our unification of clustering with stochastic neighbor embedding, our unification of spectral graph methods with contrastive learning, and our new unification of MDS with SNE. We show these connections can create new SOTA methods in image clustering and other domains further demonstrating novelty. Finally, one of the over 15 connections we document in I-Con appeared at last year’s ICLR conference. Though this paper described only a single connection, It received high reviewer scores, demonstrating the interest in this topic from the ICLR community. We note that our connections between clustering and SNE type losses are considerably more involved to prove than connections found in prior works, which is why this connection has yet to be documented.

---

> ### Author Response · Authors · 2024-11-25
>
> ## Providing Guidance on Choosing $p$ and $q$ Distributions According to I-Con
>
> Thank you for raising this point. We added a section to the supplement discussing this intuition and tried to make it a bit clearer in the main text.
>
> In summary, **I-Con provides a single unified way to view a variety of different methods in the literature as particular choices of $p$ and $q$ in Equation 1**. The I-Con framework helps reveal some of the difficult-to-see implicit distributions assumed by various methods. For example, when studying K-means, it's not easy to see that it preserves neighborhoods of uniform Gaussian variance. We elaborate on some of the key design decisions to consider when choosing $p$ and $q$.
>
> ---
>
> ### 1- **Parameterization of Learning Signal**
>
> - **Parametric**:
>   Learn a network to transform data points to representations.
>   - Use a parametric method to quickly represent new datapoints without retraining.
>   - Use this option when there are enough features in the underlying data to properly learn a representation.
>   - Ideal for datasets with sparse supervisory signals to share learning signals through network parameters.
>
> - **Nonparametric**:
>   Learn one representation per data point.
>   - Use a nonparametric method if data points are abstract and don’t contain natural features useful for mapping.
>   - Better for optimizing the loss of each individual data point.
>   - Avoid in sparse supervisory signal regimes (e.g., augmentation-based contrastive learning), as there aren’t enough links to resolve each embedding.
>
> ---
>
> ### 2- **Choice of Supervisory Signal**
>
> - **Gaussians on distances in the input space**:
>   Though common and underlying methods like K-means, with enough data, it is almost always better to use **k-neighbor distributions**, as they better capture the local topology of data.
>   This is the same intuition used to justify spectral clustering over K-means.
>
> - **K-neighbor graph distributions**:
>   If your data can naturally be put into a graph, we suggest using this instead of Gaussians in the input space.
>   - This allows the algorithm to adapt local neighborhoods to the data, rather than considering all point neighborhoods equally shaped and sized.
>   - Better aligns with the **manifold hypothesis**.
>
> - **Contrastive augmentations**:
>   When possible, add contrastive augmentations to your graph.
>   - Improves performance when quantities of interest (e.g., image class) are guaranteed to be shared between augmentations.
>
> - **General kernel smoothing techniques (e.g., random walks)**:
>   Generally improves optimization quality.
>   - Connects more points together.
>   - Sometimes mirrors geodesic distance on the manifold [Geodesics in Heat, Crane 2016].
>
> - **Debiasing**:
>   Use debiasing if you think negative pairs have a small chance of aligning positively.
>   - For a small number of classes, this parameter scales like 1/K where K is the number of classes.
>   - Improves optimization stability.
>
> ---
>
> ### 3- **Choice of Representation**
>
> Any conditional distribution on representations can be used. Consider the desired structure, such as a tree, vector, or cluster, and choose a simple, meaningful distribution for that representation.
>
> - **Discrete**:
>   Use discrete cluster-based representations if interpretability and discrete structure are important.
>
> - **Continuous Vector**:
>   Use a vector representation if generic downstream performance is a concern.
>   - Easier to optimize than discrete variants.

---

> > ### Author Response · Authors · 2024-11-28
> >
> > ## Additional Experiments: Demonstrating I-Con's Effectiveness Across Backbones and Datasets
> > We are happy to share results from additional experiments conducted to evaluate I-Con's performance on feature learning. Beyond our new feature learning experiments on CIFAR-10 and CIFAR-100, we included evaluations on STL-10 (in-domain) and Oxford-IIT Pets (out-of-domain) to support the generalization of our framework beyond clustering. Detailed descriptions of the experimental setup, along with result analyses and visualizations, are provided in the Supplement.
> >
> > | **Method**                        | **STL-10 (In-Distribution)** |                         | **Oxford-IIIT Pets (Out-of-Distribution)** |                         |
> > |-----------------------------------|-----------------------------|-------------------------|--------------------------------------------|-------------------------|
> > |                                   | Linear Probing              | KNN                     | Linear Probing                             | KNN                     |
> > | **$q_\phi$ is a Gaussian Distribution**                                                                                                                           |
> > | SimCLR [Chen et al., 2020]        | 77.71                       | 74.92                   | 74.80                                      | 71.48                   |
> > | DCL [Chuang et al., 2020]         | 78.42                       | 75.03                   | 74.41                                      | 70.22                   |
> > |                                   |                             |                         |                                            |                         |
> > | **$q_\phi$ is a Student's t-Distribution**                                                                                                                        |
> > | t-SimCLR [Hu et al., 2022]        | 85.11                       | 83.05                   | 83.49                                      | 81.41                   |
> > | DCL [Chuang et al., 2020]         | Diverges                    | Diverges                | Diverges                                   | Diverges                |
> > | **Our Debiasing $\alpha = 0.2$**  | 85.94                       | 83.15                   | 84.11                                      | 81.15                   |
> > | **Our Debiasing $\alpha = 0.4$**  | 86.13                       | **84.14**               | 84.07                                      | **84.13**               |
> > | **Our Debiasing $\alpha = 0.6$**  | **87.18**                   | 83.58                   | **84.51**                                  | 83.04                   |
> >
> > In summary, the results support that methods focused on distribution improvements—such as debiasing and replacing Gaussian kernels with Cauchy kernels—significantly enhance feature quality across datasets evident both in linear probe accuracy and KNN accuracy.

---

> > > ### Comment · Reviewer_xJUE · 2024-11-30
> > >
> > > Thank you for your response, which addresses most of my concerns. I have updated my score from 5 to 6.
> > >
> > > Here are some addition suggestions. The authors may take into consideration and extend this work to a journal like TPAMI:
> > > - As previously mentioned, drawing inspiration from other domains is a natural part of the research process. On one hand, the approach to selecting the mechanism based on the I-Con framework is valuable, and I appreciate that the authors have addressed this part well. On the other hand, it would be even more valuable to propose a new mechanism based on this theorem, which appears to be an area not yet explored in the current work.
> > > - The experimental section primarily focuses on ablation studies rather than comparisons with state-of-the-art methods. Combining the idea of proposing a new mechanism with a comparison to the latest methods could strengthen the contribution and provide a clearer perspective on the improvements achieved.

---

> > > > ### Author Response · Authors · 2024-12-01
> > > > **Thank you!**
> > > >
> > > > Thank you for your thoughtful feedback, suggestions, and for raising your score!

---

### Author Response · Authors · 2024-11-25
**General Response**

We appreciate the reviewers thoughtful and earnest feedback and have used this feedback to improve the work. We also appreciated that reviewers though our paper had “practical significance” (xJUE), “very well written and well organized” (AnKo), “novel” (AnKo), and showed “significant improvement” (FvTp). We have provided a global summary of changes in this revision, as well as a specific rebuttal to each reviewer. We have also uploaded a revised version of the paper with many requested additions and fixes.
## 1- New Series of Experiments Demonstrating that I-Con improves Feature Learning and works on different backbones and datasets.

In a new series of experiments we show that I-Con not only improves clusterers, but can improve contrastive feature learners. As requested by reviewers this experiment also demonstrates that I-Con works on an additional backbone and datasets. We follow the contrastive feature learning experiment put forth by Hu et al. at ICLR 2023. In particular we train a ResNet34 network contrastively on CIFAR10 and evaluate its linear probe and KNN performance on both CIFAR10 and CIFAR100. We include a detailed description of this experimental setting and its results in the Supplement. In summary, we show that our distribution improvement methods like debiasing and replacing gaussian kernels with cauchy kernels dramatically improves feature quality both in terms of linear probe accuracy and KNN accuracy.

| **Method**                          | **CIFAR10 (in distribution)** |                        | **CIFAR100 (out of distribution)** |                     |
|-------------------------------------|-------------------------------|------------------------|-------------------------------------|---------------------|
|                                     | Linear Probing               | KNN                    | Linear Probing                      | KNN                 |
| **$q_\phi$ is a Gaussian Distribution**                                                                                                 |
| SimCLR [Chen et al., 2020]          | 77.79                        | 80.02                  | 31.82                               | 40.27               |
| DCL [Chuang et al., 2020]           | 78.32                        | 83.11                  | 32.44                               | 42.10               |
| **Our Debiasing $\alpha = 0.2$**    | 79.50                        | 84.07                  | 32.53                               | 43.19               |
| **Our Debiasing $\alpha = 0.4$**    | 79.07                        | 85.06                  | 32.53                               | **43.29**           |
| **Our Debiasing $\alpha = 0.6$**    | 79.32                        | 85.90                  | 30.67                               | 29.79               |
|-------------------------------------|-------------------------------|------------------------|-------------------------------------|---------------------|
| **$q_\phi$ is a Student's t-distribution**                                                                                             |
| t-SimCLR [Hu et al., 2022]          | 90.97                        | 88.14                  | 38.96                               | 30.75               |
| DCL [Chuang et al., 2020]           | Diverges                     | Diverges               | Diverges                            | Diverges            |
| **Our Debiasing $\alpha = 0.2$**    | 91.31                        | 88.34                  | 41.62                               | 32.88               |
| **Our Debiasing $\alpha = 0.4$**    | 92.70                        | 88.50                  | **41.98**                           | 34.26               |
| **Our Debiasing $\alpha = 0.6$**    | **92.86**                    | **88.92**              | 38.92                               | 32.51               |


## 2- Proving that MDS, PCA (Linear Autoencoders), Masked Language Modeling, Spectral Clustering, Mocov3, and Triplet Loss are unified by I-Con

We introduce new theorems that unify 6 additional methods with the I-Con framework. In particular we show that MDS and SNE can be unified in the limit of large gaussian scales, which is a novel theoretical result first proved in this work. As requested by reviewer iHot, this result allows us to integrate linear autoencoders (PCA) into I-Con.  As requested by reviewer iHot we also show masked language modeling can be considered in the I-Con framework. Intuitively this  is a simple corollary of our existing proof connecting Cross Entropy with I-Con, except replacing class embeddings with embeddings for masked tokens. Finally, we show that the commonly used triplet loss for contrastive learning emerges as in the limit of small width gaussians in I-Cons InfoNCE connection. These additional proofs can be found in the supplement.

---

> ### Author Response · Authors · 2024-11-25
>
> ## 3- Connecting I-Con with Variational Bayes and Mutual Information Maximization
> We add sections relating I-Con to both Mutual Information maximization and Variational Bayesian methods, as requested by reviewer AnKo. Intuitively speaking, I-Con is linked to Mutual Information Maximization through a variational lower bound. I-Con is theoretically related to Variational Bayesian Methods as one view can represent hidden variables $h$ and the other observed variables $x$. We include detailed analysis and discussion of these intuitions in 2 new supplemental sections.
>
> ## 4- Providing guidance on choosing conditional distributions $p$ and $q$ and hyperparameters
>
> As requested, we introduce a new section of the paper that details the different intuitions we have discovered using the I-Con framework. In particular many of these intuitions are based around choosing neighborhood distributions $p$ and $q$. These distributions should be carefully considered depending on the problem at hand, available information, and the desired output of the unsupervised learning method. However we highlight several generic probability distribution modification techniques that have generic interpretations as debiasers, or distribution smoothers. We show these general transformations yield both state of the art clusterers, and improved contrastive representation learners.
>
> ## 5-Improving motivation of Unifying Framework
> We add a section further motivating the I-Con framework and detailing some of the things a unified framework provides. As noted by reviewer xJUE, this unification “not only provides a deeper understanding of these methods but also opens up the possibility of creating new methods by mixing and matching components.” We explicitly used this property to discover new improvements to both clustering and representation learners.  In short, I-Con acts like a periodic table of machine learning losses. With this periodic table we can more clearly see the implicit assumptions of each method by breaking down modern ML losses into more simple components: pairwise conditional distributions $p$ and $q$.
>
> One particular example of how this opens new possibilities is with our generalized debiasing operation. Through our experiments we show adding a slight constant linkage between datapoints improves both stability and performance across clustering and feature learning. Unlike prior art, which only applies to specific feature learners, our debiasers can improve clusterers, feature learners, spectral graph methods, and dimensionality reducers.
>
> Finally it allows us to discover novel theoretical connections by compositionally exploring the space, and considering limiting conditions. We use I-Con to help derive a novel theoretical equivalences between K-Means and constrastive learning (initial work) and between MDS, PCA, and SNE (In this rebuttal and in the new supplement). Reviewer xJUE mentioned that transferring ideas between methods is standard in research, but in our view it becomes much simpler to do this if you know methods are equivalent. Previously, it might not be clear how exactly to translate an insight like changing gaussian distributions to Cauchy distributions in the upgrade from SNE to T-SNE has any effect on clustering or representation learning. In I-Con it becomes clear to see that similarly softening clustering and representation learning distributions can improve performance and debias representations.
>
> ## 6- Improving experimental robustness with multiple seeds
>
> As requested by reviewer FvTp we report standard errors on our I-Con result in Table 2 to demonstrate that the improvements are indeed statistically observable.
>
> ## 7- Fixes, Correcting Typos, and other Clarifications
> We have corrected several typos raised by reviewers, and tried to provide more elaboration on tpoints requested by reviewers. Finally we add a variety of sections into the supplemental material that provide detailed explanations and answers to many questions raised by reviewers including
> 1) Detailing the motivation of I-Con
> 2) Connecting I-Con with other literature reviews
> 3) Explaining of how to select distributions $p$ and $q$.
> 4) Explaining the relation between I-Con and Variational Bayes
> 5) Explaining the relation between I-Con and classical MLE inference
> 6) Clarifying the role of hyperparamter tuning and clarifying our claim that I-Con is self-balancing

---

> ### Author Response · Authors · 2024-11-28
>
> ## Additional Experiments: Demonstrating I-Con's Effectiveness Across Backbones and Datasets
> We are happy to share results from additional experiments conducted to evaluate I-Con's performance on feature learning. Beyond our new feature learning experiments on CIFAR-10 and CIFAR-100, we included evaluations on STL-10 (in-domain) and Oxford-IIT Pets (out-of-domain). Detailed descriptions of the experimental setup, along with result analyses and visualizations, are provided in the Supplement.
>
> | **Method**                        | **STL-10 (In-Distribution)** |                         | **Oxford-IIIT Pets (Out-of-Distribution)** |                         |
> |-----------------------------------|-----------------------------|-------------------------|--------------------------------------------|-------------------------|
> |                                   | Linear Probing              | KNN                     | Linear Probing                             | KNN                     |
> | **$q_\phi$ is a Gaussian Distribution**                                                                                                                           |
> | SimCLR [Chen et al., 2020]        | 77.71                       | 74.92                   | 74.80                                      | 71.48                   |
> | DCL [Chuang et al., 2020]         | 78.42                       | 75.03                   | 74.41                                      | 70.22                   |
> |                                   |                             |                         |                                            |                         |
> | **$q_\phi$ is a Student's t-Distribution**                                                                                                                        |
> | t-SimCLR [Hu et al., 2022]        | 85.11                       | 83.05                   | 83.49                                      | 81.41                   |
> | DCL [Chuang et al., 2020]         | Diverges                    | Diverges                | Diverges                                   | Diverges                |
> | **Our Debiasing $\alpha = 0.2$**  | 85.94                       | 83.15                   | 84.11                                      | 81.15                   |
> | **Our Debiasing $\alpha = 0.4$**  | 86.13                       | **84.14**               | 84.07                                      | **84.13**               |
> | **Our Debiasing $\alpha = 0.6$**  | **87.18**                   | 83.58                   | **84.51**                                  | 83.04                   |
>
> In summary, the results support that methods focused on distribution improvements—such as debiasing and replacing Gaussian kernels with Cauchy kernels—significantly enhance feature quality across datasets. This improvement is evident both in linear probe accuracy and KNN accuracy.

---

### Public Comment · ~Aditya_Ravuri1 · 2025-04-29
**Questions to authors**

Hello! I was just made aware of this work, and I'm really glad that there is increasingly more interest in probabilistically unifying representation learning algorithms! I have a few questions to the authors. (Also, this is based on an initial read, massive apologies in advance if I've misunderstood certain ideas!)

Very briefly:
  * what random variables are the distribution $p(j|i)$ over?
  * how does this compare to prior work in this direction?

To elaborate a bit further:
 1. There has been prior work in this direction, that achieves a very similar taxonomical end goal as yours. Would love to see a discussion of these and it'd also help one understand how your framework compares, and provide alternative routes to derive those methods! It's super interesting to see connections from other point of views.

    One such work is [1]. In this work, we also unify most classical dimensionality reduction methods as KL minimising algorithms, including UMAP, t-SNE, (dual-P)PCA, MDS, etc. In [2], we offer a different take on UMAP and t-SNE as MAP algorithms, but due to the results from [1], these can be interpreted variationally too.

    In a nutshell, in [1], we show that SNE, t-SNE and UMAP correspond to KL(q||p) minimisation algorithms, with Categorical/Bernoulli distributions on the data-pairs (the random variable in question corresponds to a slice of an adjacency matrix). [6] also find very similar results as these. Also there's been some really cool work trying to understand the nature of UMAP and t-SNE's true loss functions in [7, 8].

    Moreover, algorithms such as (dual-probabilistic) PCA, MDS, kPCA, Laplacian Eigenmaps, Locally Linear Embedding, etc. are all cases in ProbDR of a KL minimisation on a covariance/precision matrix (which is the random variable), endowed with a Wishart distribution [1].

    I realise that a KL can be defined in many different ways, over different _things_ (random variables), so perhaps what we talk about are approximations / different representations of the same thing, and a lot of my questions come from viewing these methods from a ProbDR point of view.

    **A question to the authors:** what is the random variable that $p(j|i)$ is defined on? Is this perhaps a discrete matrix-valued random variable that corresponds to data-pair similarity, or are the real-valued elements of $\mathcal X$ endowed with the distributions in Table 1?

 2. Questions on (t-)SNE:
     (Continuing from point 1) Table 1 suggests that the real-valued data points are endowed with a Gaussian distribution, but I struggle to understand Theorem 1 from this light. In theorem 1, it's easier to see p and q as Categorical distributions over a discrete choice (is data point j similar to data point i).

     Apologies if this seems pedantic, but it's I think important to iron this out. I've disagreed with the use of the word ``distribution'' in the original SNE and t-SNE papers, and prefer the terms 'Gaussian kernels' or 't/Cauchy kernels' for the choice of probabilities over categorical random variables, because I do not understand how such probabilities can be derived (and over what) assuming Gaussian data.

     The closest perhaps is Gaussian mixture model. Assuming that each data point is drawn one point at a time from a gaussian mixture:
     $$x_i | z_{\not i} \sim \mathcal N(\sum_{j \neq i} z_j * x_j, \sigma^2),$$
     $$ z \sim Categorical(\mathbf{1}/(n-1)), $$
     Then, I think that the posterior over the latent discrete variables $z$ would be such that:
     $$ \mathbb P(z_j=1 | x) = \exp(-0.5\sigma^{-1}||x_i - x_j ||^2) / \sum_{k \neq i} \exp(-0.5\sigma^{-1}\|x_i - x_k \|^2)$$
     In this case, I'd describe the data points as conditionally drawn from a mixture of Gaussians, and note that (as far as I can see), this does not mean that x_i jointly or even marginally have a Gaussian distribution.

     I believe that this is important, because if the data did jointly or even marginally have a Gaussian distribution, it'd be easy to relate them to Gaussian Process Latent Variable Models and PCA, and I've not seen an easy way to do this.

     Again, please do let me know if I'm missing something as I've always wondered why the original SNE and t-SNE papers refer to the ``distribution'' over the latents/data as Gaussian/Cauchy(t-) distributed.

     Taking a different route, if a random variable $y_i$ were defined as $y_i = \mathcal{I} (\|y_i - y_j \|^2 < \epsilon)$, the probabilities for $y_i$ do depend on the Euclidean distances between the data points, but these formulations are quite ugly and don't result in the nice form seen in t-SNE (here, I use the fact that the squared distances are marginally Gamma distributed). I apologise if this is easy to see, but it's not obvious to me what transformation applied to the real valued gaussian data to a set of categorical random variables results in probabilities of the kind that appear in SNE and t-SNE.

(Continued in the comment)

---

> ### Public Comment · ~Aditya_Ravuri1 · 2025-04-29
> **Questions continued**
>
> 3. Questions on PCA:
>      It's unclear to me what the PCA objective is taken to be - the last line of Theorem 2, $\mathcal{L} = -2 Var(X)$ seems to not depend on the projection matrix? Say that we try to find a low dimensional representation $\mathbf{Z} \in \mathbb{R}^{n, l}$, which is related to the high-dimensional data $\mathbf{X} \in \mathbb{R}^{n, d}$ in a linear way: $\mathbf{X} = \mathbf{ZW}$. The objective typically involves $tr(\mathbf{X}^T\mathbf{X} (\mathbf{W}^T\mathbf{W})^{-1})$ - this can be seen through a least squares approach, or through a probabilistic perspective as in [3]. Inference for $\mathbf{Z}$ intead of $\mathbf{W}$ can be done instead in a very similar way [4].
>
>      Maximum likelihood/MAP interpretations have a KL minimisation interpretation, but note here that _*both distributions p and q*_ are either Gaussian (if we consider the data directly [4, Sec 2.6.1]) or both Wishart (if we consider the covariance matrix $\mathbf{XX^T}$ or $X^TX$) [1].
>
>      If I'm understanding correctly, there is a discrete random variable considered and not the data here directly, and you show that KL minimisation here recovers PCA? That's super cool but I feel like there are some details missing, e.g. in Theorem 2, regarding the objective.
>
>  4. On SimCLR:
>      I'm not very familiar with this work, but in [5], SimCLR was shown to be a KL minimisation algorithm with a von-Mises-Fisher distribution on the latents (distribution "p"), and the variational constraint ("q") is degenerate. I think there's again a difference in the random variable on which a distribution is placed, but it'd be amazing if the authors could verify consistency between these interpretations (as it's suggested in this work that the distribution is Gaussian instead of vMF? These are similar, in the latter case, the embeddings just have unit norm).
>
> ### References
>
> [1] Dimensionality Reduction as Probabilistic Inference, Ravuri et al. (2023). [Accessible here.](https://arxiv.org/pdf/2304.07658)
>
> [2] Towards One Model for Dimensionality Reduction, Ravuri et al. (2025). [Accessible here.](https://openreview.net/pdf?id=HMJod9Jg8r)
>
> [3] Probabilistic Principal Component Analysis, Tipping and Bishop (1999).
>
> [4] Probabilistic Non-linear Principal Component Analysis with Gaussian Process Latent Variable Models, Lawrence (2005).
>
> [5] Representation Uncertainty in Self-Supervised Learning as Variational Inference, Nakamura et al. (2023)
>
> [6] A Probabilistic Graph Coupling View of Dimension Reduction, Van Assel et al. (2022)
>
> [7] On UMAP's True Loss Function, Damrich et al. (2021)
>
> [8] From t-SNE to UMAP with contrastive learning, Damrich et al. (2022)

---

### Meta-Review · Area_Chair_MW6R · 2024-12-26

**Metareview:**

The paper presents a framework that unifies various representation  learning methods,including clustering, contrastive learning, dimensionality reduction, and supervised learning, as a minimization of the KL divergence between two distributions, this covers 15 known techniques.  Authors provide a method for distribution improvement methods such as debiasing and replacing gaussian kernels with cauchy kernels which improve feature quality both in terms of linear probe accuracy and KNN accuracy.Results on imagenet are provided , giving 8% improvement on previous methods.

**Additional Comments On Reviewer Discussion:**

Authors and reviewers discussed the paper which resulted in  improvement of the manuscript.

Reviewers agreed that the contribution of the paper though not novel,  it is insightful to  the ICLR community for designing new losses. Please integrate all feedback of the reviewers that was not added to the manuscript.

---

### Decision · Program_Chairs · 2025-01-22

Accept (Poster)